# Rainfall events with shallow landslides in the Entella catchment, Liguria, Northern Italy

Anna Roccati[1], Francesco Faccini[2], Fabio Luino[1], Laura Turconi[1], Fausto Guzzetti[3]

[1] Istituto di Ricerca per la Protezione Idrogeologica, Consiglio Nazionale delle Ricerche, Strada delle Cacce 73, 10135 Torino, Italy

[2] Dipartimento di Scienze della Terra, dell'Ambiente e della Vita, Università di Genova, Corso Europa 26, 16132 Genova, Italy

[3] Istituto di Ricerca per la Protezione Idrogeologica, Consiglio Nazionale delle Ricerche, Via Madonna Alta 126, 06128 Perugia, Italy

*Correspondence to*: Anna Roccati (anna.roccati@fastwebnet.it)

**Abstract.** In the recent decades, the Entella River basin, in the Liguria Apennines, Northern Italy, was hit by numerous intense rainfall events that triggered shallow landslides and earth flows, causing casualties and extensive damage. We analysed landslides information obtained from different sources and rainfall data recorded in the period 2002–2016 by rain gauges scattered in the catchment, to identify the event rainfall duration, $D$ (in h), and rainfall intensity, $I$ (in mm h$^{-1}$), that presumably caused the landslide events. Rainfall-induced landslides affected all the catchment area, but were most frequent and abundant in the central part, where the three most severe events hit on 23-24 November 2002, 21–22 October 2013, and 10-11 November 2014. Examining the timing and location of the slope failures, we found that the rainfall-induced landslides occurred primarily at the same time or within six hours from the maximum peak rainfall intensity, and at or near the geographical location where the rainfall intensity was largest. Failures involved mainly forested and natural surfaces, and secondarily cultivated and terraced slopes, with different level of maintenance. Man-made structures characterize frequently the landslide source areas. Adopting a Frequentist approach, we define the event rainfall intensity–event duration ID, threshold for the possible initiation of shallow landslides and hyper-concentrated flows in the Entella River basin. The threshold is lower than most of the curves proposed in the literature for similar mountain catchments, local areas and single regions in Italy. The result suggests a high susceptibility to rainfall-induced shallow landslides of the Entella catchment due to its high-relief topography, geological and geomorphological settings, meteorological and rainfall conditions and human interference. Analysis of the antecedent rainfall conditions for different periods, from 3 to 15 days, revealed that the antecedent rainfall did not play a significant role in the initiation of landslides in the Entella catchment. We expect that our findings will be useful in regional to local landslides early warning systems, and for land-planning aimed at reducing landslides risk in the study area.

# 1 Introduction

The study concerning the rainfall thresholds able to trigger shallow landslides is certainly one of the most studied aspects in the geomorphological field in the last decades. The idea of being able to predict the triggering of a mass movements as a function of recorded rainfall has greatly fascinated ranks of researchers all over the world. The study can be exhaustive only if the researcher has precise rainfall data and detailed information about the location, altitude and the time of the landslide trigger.

Many papers have been published about the sort of statistical analysis of the relationship between rainfall and the occurrence of mass-movements (Campbell, 1975; Caine, 1980; Govi and Sorzana, 1980; Moser and Hohensim, 1983; Wieczorek and Sarmiento, 1983; Brand et al. 1984; Govi et al., 1985; Cannon and Ellen, 1985; Crozier, 1986; Wieczorek, 1987) since the beginning of the eighties. And they continued with many other studies also in the following decade (Kim et al., 1991; Page et al., 1993; Ceriani et al., 1994; Premchitt et al., 1994; Crozier, 1997; Glade, 1997; Crosta, 1998).

Italy proves to be an excellent experimental field. Every year, in particular during the summer, there are thousands of shallow landslides due to short but heavy rainfall: they can cause serious social and economic damage (Wasowski, 1998; Alpert et al., 2002; D'Amato Avanzi et al., 2012; Luino, 2005; Giannecchini, 2006; Del Ventisette et al., 2012; Trigila et al., 2018).

Since 1970, the Liguria region (northern Italy), one of the Italian areas where landslide risk is the highest (Guzzetti, 2000; Salvati et al., 2010), suffered from rainfall events characterized by flash floods, shallow landslides, large scale complex landslides and earth flows along small creeks (Faccini et al 2015c). Since the early 1990s, the Liguria region seems to be suffering a sort of resurgence of these events, more and more characterized by short duration and intense rainfall that have triggered damaging landslides (Guzzetti et al., 2004; Faccini et al., 2005; 2015a; 2015b; 2017; Giannecchini et al., 2010; 2015; Silvestro et al., 2012; 2016; Cevasco et al., 2015; 2017; D'Amato Avanzi et al., 2015), chiefly where urban planning and land management were poor or ineffective, resulting in an increased vulnerability to geo-hydrological (landslides and flood) hazards (Bartolini et al., 2014). Globally, since 1970 the rainfall induced landslide events which have claimed 16 lives and have caused widespread and extensive damage to private and public properties, structures and infrastructures, and business.

The acknowledge of the amount of precipitation needed to trigger shallow landslides and earth flows is essential for the implementation of an operational landslide warning system (Aleotti, 2004; Baum and Godt, 2010; Tiranti and Rabuffetti, 2010; Martelloni et al., 2012; Mathew et al., 2014). Definition of rainfall thresholds is the key issue for the prediction of rainfall-induced landslides. In literature, various methods have been proposed to define the rainfall condition that are likely to trigger slope failures. Physically based approaches depend on the understanding of the complex physical processes that control slope stability (Montgomery and Dietrich, 1994; Wilson and Wieckzorek, 1995; Iverson 2000; Crosta and Frattini, 2003; Segoni, 2009): they are elaborate to apply because of the difficulty of defining the exact spatial and temporal variation of the many factors involved, particularly on large regions (e.g., effects of vegetation, rainfall variation, mechanic and

hydraulic properties of both bedrock and soils). Empirical approaches rely on the definition of empirical or statistical rainfall thresholds through the analysis of the rainfall conditions that in the past triggered landslides (Caine, 1980; Wieczorek, 1996; Aleotti, 2004; Guzzetti et al., 2008; Brunetti et al., 2010; Saito et al., 2010; Palladino et al., 2017; Peruccacci et al., 2017); they differ in rainfall data adopted (e.g., intensity and duration of precipitation, cumulative event rainfall, antecedent rainfall, etc.) and geographical extent of their validity (e.g., local, regional, national, global).

Here we discussed the rainfall events that have triggered shallow landslides in the Entella River basin, in eastern Liguria, in the 15-year period 2002–2016. During this period, the study area was one of the most frequently affected catchments, with a total of 29 damaging rainfall events and 94 known shallow landslides that have caused 2 fatalities. We use rainfall and landslide information of a subset of 17 events, for which location and timing of the failures are exactly known, to define the rainfall thresholds for the possible initiation of shallow landslides in the Entella catchment. In addition, we investigate the correlation between landslides and land cover and the possible role of the antecedent rainfall in slope failures.

The paper is organized as follows: i) description of the general setting of the Entella catchment (Section 2); ii) description of the sources and criteria used for the compilation of the catalogue of rainfall events that triggered shallow landslides and the method adopted to define the rainfall thresholds (Section 3) ; iii) focus on the three largest and damaging rainfall events (23-24 November 2002, 21-22 October 2013 and 10-11 November 2014) that affected the study area (Section 4); iv) rainfall thresholds for the possible initiation of shallow landslides in the Entella catchment and results of the rainfall and landslides analysis (Section 5); v) discussion of our findings, including a comparison with similar, published curves and the correlation with the land-use and human interference (Section 6) and vi) summary of the results of the study and its possible implementation in a landslide warning system and land planning (Section 7).

## 2 Study area

The Entella River basin extends for 375 km$^2$ in the Tyrrhenian sector of the Liguria Apennines, north-western Italy (Fig. 1) and encompasses the Lavagna (160 km$^2$), Sturla (130 km$^2$), and Graveglia (63 km$^2$) tributaries (Fig. 1A). In the area crop out different rock types, Jurassic to Paleocene in age, arranged in a complex geological setting characterized by the presence of multiple sets of tectonic discontinuities (Geological Service of Italy, 1968; Liguria Region, 2005) (Fig. 1B). Shale, limestone, heterogeneous marl, slate and sandstone crop out in the Lavagna and lower Sturla valleys, whereas in the Graveglia valley crop out an ophiolitic sequence, encompassing serpentine, gabbro, basalt, ophiolite breccia, chert, grey limestone and shale. In the upper part of the Sturla valley, along the Tyrrhenian–Adriatic divide, crop out marly limestone, marl, and other heterogeneous and chaotic rocks. Pliocene and Quaternary deposits cover extensively the slopes and the valley floors. The morphology of the catchment is controlled by the geological and the structural settings. Except for narrow and mostly flat plains in the Lavagna, Graveglia and Sturla valleys, steep and very steep slopes characterize the catchment, some of which host large and deep-seated landslides. Different types of landslides generally characterize the Entella catchment, with different state of activity, from active to dormant and stabilized: i) shallow landslides, slow-moving and

rapid to extremely rapid earth flows (Hungr et al., 2001) and rock falls on steep slopes (in particular along the roads), triggered by very short and intense rainfalls, and ii) ancient slow large deep-seated gravitational slope deformations, particularly at the head of the valleys, reactivated during prolonged precipitations. Landslides are widely distributed and involve mainly debris cover from few centimeters to 3-4 meters thick, and portions of the fractured and weathered bedrock (Faccini et al., 2002, 2017; Cevasco et al, 2017). Slope failures affect areas from few square meters to several square kilometers (DSGSD) and involve volumes ranging from few to thousands of cubic meters. Forests cover more than 70% and urban areas less than 3% of the catchment; the later chiefly along the main valley floors (Liguria Region, 2000). The remaining part is covered by olive groves, chestnut woods and fruit orchards on well-maintained or abandoned terraces, and by grassland and grazing land. Shrubs and outcropping rocks are common at high elevation

Climate is Mediterranean, modulated by the local morphological and orographic conditions. The presence of valleys oriented E-W (Lavagna and lower Graveglia valleys) and NNE-SSW (Sturla and upper Graveglia valleys) facilitates the channelling of air masses and meteorological disturbances pushed landward by southern marine winds. Mean annual precipitation (MAP) averages 1800 mm, and ranges from 1130 mm near the basin outlet, to more than 2300 mm in the upper Lavagna and Sturla valleys. Precipitation is most abundant from October to November, and in February. The Apennines range, exceeding locally 1700 m in elevation, runs parallel to the cost of the Liguria Sea, and acts as a barrier for the low-pressure systems that characterized the Gulf of Genoa, primarily in the autumn and winter periods (Anagnostopoulou et al., 2006): these typical Mediterranean cyclones of orographic origin, called "Genoa Low", result in thunderstorms with intense and very intense rainfall that cause widespread and damaging ground effects, both on the slopes and along the drainage pattern, in the Ligurian and Northern Toscana Regions (Brandolini et al., 2012; Sacchini et al., 2012; Silvestro et al., 2012, 2015).

**3 Methods**

To define the rainfall conditions that may trigger shallow landslides in our study area, we considered 29 rainfall events occurred in the Entella catchment in the 15-year period 2002–2016 that have caused landslides (Table 1). All the events caused damage (Fig. 2), and one of them (10-11 November 2014) resulted in loss of lives. For each event, we collected landslide and rainfall information. The landslide information included (i) the location and number of the event landslides, (ii) the time of occurrence of the slope failures, and (iii) the consequences of the landslides (i.e., type of damage, casualties). We obtained the landslide information from different sources, including scientific papers, technical and event reports, damage reports, and catalogues compiled by regional and local authorities, archives of local municipalities, newspaper articles, and interviews to local inhabitants. We obtained rainfall measurements from 15 rain gauges in the Entella catchment (Fig. 1, Table 2). The rain gauges operated for different periods and show gaps in their records, including a gap common to all gauges in the period 1999–2001. For each rain gauge, we calculated the mean annual precipitation (MAP) for the entire measurement record.

Among the 29 rainfall events, we collected useful information for 16 events. Using this subset of landslide and rainfall information, we identified 34 rainfall conditions associated to known landslides for which the time and location of the slope failures were known with sufficient geographical and temporal accuracy (Table 3). For each rainfall condition, we estimated the cumulated event rainfall, $E$ (in mm), the event rainfall duration, $D$ (in h), the rainfall intensity, $I$ (in mm h$^{-1}$) for 1, 3, 6, 12 and 24-hour periods, and the average rainfall intensity for the rainfall event, $\hat{I}$ (mm h$^{-1}$) (Table 3). For the purpose, following Peruccacci et al. (2012), for each known landslide or cluster of landslides, we selected a representative rain gauge, considering (i) the geographic distance between the rain gauge and the landslide, or the geometric centre of a cluster of landslides, (ii) the elevation of the rain gauge, compared to the elevation of the landslide(s), and (iii) the location of the rain gauge with respect to the geographical and morphological settings. For 82% of the landslides, the representative rain gauge was the one closest to the landslide or the cluster of landslides. The rainfall duration ($D$, in hour) was determined measuring the period between the end-time of the rainfall event, set to coincide with the time of the landslide(s), and the start-time of the rainfall event, set to coincide with the time when the rain started in the rainfall record. For landslides for which the rainfall end-time s known accurately, the end-time coincides with the time of the last rainfall measurement in the hour when the slope failure occurred. For landslides for which only the date of occurrence is known, the rainfall end-time is taken to coincide with the time of the last rainfall measurement in the day when the slope failure occurred.

When the accurate identification of the rainfall start-time was problematic in the rainfall record, we considered a minimum period without rain ("dry period") to separate two subsequent rainfall events. To account for the seasonal variability, we considered a dry period of 12 hours between April and September, and a dry period of 24 hours between October and March.

For each rainfall event, the corresponding event rainfall intensity ($I$, in mm h$^{-1}$) was calculated dividing the cumulated event rainfall in the considered period ($E$, in mm) by the length of the rainfall period ($D$, in h). For the rainfall events listed in Table 1 for which the date of occurrence of the landslides was unknown, the reconstruction of the rainfall event was not possible; and the events were not used to define the ID rainfall threshold.

To define the rainfall duration–rainfall intensity threshold, we adopted the statistical Frequentist approach proposed by Brunetti et al. (2010). We plotted the 34 event rainfall duration–rainfall intensity ($D$, $I$) conditions that have resulted in shallow landslides and earth flows in the period 2002–2016, and we found the best-fit to the cloud of empirical ($D$, $I$) data points, adopting a power law model. To avoid problems associated with the fitting of data spanning multiple orders of magnitude, we log-transformed the empirical data, and we fitted the distribution of rainfall conditions (log($D$), log($I$)) that resulted in landslides with a linear equation,

$$\log(I) = \log(\alpha) - \beta \log(D) \tag{1}$$

which is entirely equivalent to the power law curve

$$I = \alpha D^{-\beta} \tag{2}$$

in linear coordinates commonly adopted to represent ID thresholds in the literature (Guzzetti et al., 2007; 2008), where $I$ is the rainfall intensity (mm hr$^{-1}$), $D$ is the duration of the rainfall event (h), $\alpha$ is the intercept, and $\beta$ defines the slope of the power law curve.

Next, for each event ($D$, $I$) we calculated the difference $\delta(D)$ between the logarithm of the event intensity $\log[I(D)]$ and the corresponding intensity value of the fit $\log[I_f(D)]$, $\delta(D) = \log[I(D)] -$ fit $\log[I_f(D)]$. Then we estimated the probability density of the distribution of $\delta$ and model the distribution through least square fitting using the Gaussian function:

$$G(\delta) = \frac{1}{\sqrt{2\pi\sigma^2}} * \exp\left\{-\frac{[\delta-\mu]^2}{2\sigma^2}\right\}$$ (3)

where $\mu$ is the mean value, and $\sigma$ is the standard deviation. Lastly, the threshold corresponding to the 5% exceeding probability, is defined, based on the fitted distribution of $\delta(D)$.

Peruccacci et al. (2012) have argued that empirical rainfall thresholds for possible landslide occurrence are inherently affected by uncertainty, and that the uncertainty needs to be quantified. In an attempt to define the uncertainty associated to our threshold, we calculated the mean values $\bar{\alpha}$ and $\bar{\beta}$ and the associated uncertainty $\Delta\alpha$ and $\Delta\beta$ in terms of standard

deviation of the $\alpha$, $\beta$ values obtained after sampling the $D$, $I$ conditions considered to define the threshold. First, out of the N($D$, $I$) population, we selected a single ($D$, $I$) value, and we calculated the $\alpha$ and $\beta$ values using the remaining (N-1) population of (D,I) value. When the selected ($D$, $I$) values was returned to the population N, a second ($D$, $I$) value was selected, and new $\alpha$, $\beta$ values calculated. We repeated the procedure for all the n=34 (D,I) conditions, and calculated the standard deviation of the $\alpha$ and the $\gamma$ values.

We compare the obtained curve to similar ID empirical thresholds proposed in the literature for similar mountain catchments, local areas or single regions in Italy to investigate the influence of the local physiographical (e.g., meteorological, morphological, lithological, hydrological) and land cover setting on the rainfall thresholds.

To investigate the possible role of the antecedent rainfall conditions in the initiation of the rainfall-induced landslides in the Entella catchment, for each rainfall event in the 15-year period 2002–2016, we confronted the MAP-normalized cumulated

event rainfall $E_{MAP}$, to the MAP-normalized antecedent rainfall, $A_{(d)MAP}$ for four antecedent periods − $A_{(3)MAP}$, $A_{(5)MAP}$, $A_{(15)MAP}$ and $A_{(30)MAP}$ − for periods of 3, 5, 15, and 30 days before the rainfall events. Each rainfall record was normalized to the MAP of the same rain gauge used to define the $D$, $I$ rainfall conditions.

## 4 Rainfall events and shallow landslides

The most severe rainfall events, in terms of abundance of shallow landslides and extent of damage, that affected the Entella

catchment in the considered period 2002−2016 are those occurred on 23-24 November 2002, 21-22 October 2013 and 10-11 November 2014. In the next sections, a briefly focus of them is carried out to illustrate the peculiar meteorological conditions that caused intense and damaging precipitations in the Entella catchment, and the rainfall conditions and their ground-effects.

## 4.1 The 23-24 November 2002 rainfall event

On 24 November 2002, a short and very intense rainfall event hit the medium-to-lower part of the Lavagna valley (Fig. 1). The convergence between a cold northern draft and wet Mediterranean currents generated a low-pressure system in the Tigullio Gulf that moved to the medium-lower Lavagna valley. The very intense rainstorm hit an area of about 50 km$^2$ for a period of 1–2 hours, before moving towards NNE. The event started in the afternoon on 23 November, and continued in the late morning on 24 November, with a second and more intense rainfall burst. Maximum cumulated rainfall values were

measured (i) at the San Martino del Monte rain gauge, the closest to the geometric centre of clusters of event-triggered landslides, that recorded 79.4 mm of rain in one hour, and (ii) at the Piana di Soglio rain gauge, that recorded 97.2 mm in 3h, and 104.6 mm in 6h. The cumulated rainfall for the event exceeded 157.0 mm (Fig. 3a). The 1-hr maximum intensity reached 93.0 mm between 12:00 and 13:00 UTC at the Chiavari rain gauge, with the most intense rainfall measured between 09:00 and 13:00 UTC on 24 November, corresponding to the most intense phase of the event.

Figure 4a portrays the cumulated event rainfall between 08:00 and 14:00 UTC on 24 November 2002, and shows that the maximum rainfall occurred along a NNE-SSW trending line, from the medium-lower Lavagna valley (Piana di Soglio, 104.6 mm) to the upper part of the Sturla valley (Tigliolo, 124.8 mm). The rainfall field decreased gradually towards SE (San Martino del Monte, 80.0 mm; San Michele, 76.0 mm) and towards NW, with a minimum value in the upper part of the Lavagna valley (Neirone, 14 mm).

The rainfall event triggered numerous shallow landslides and earth flows, which caused serious damage chiefly to the road network (Faccini et al., 2005). Based on evidences provided by local inhabitants, landslides occurred in the 30-minute period between 14:00 and 14:30 LT (13:00–13:30 UTC), following the most intense phase of the rainfall event, and within 1 to 4 hours from the hourly maximum peak rainfall intensity recorded by the San Martino del Monte rain gauge in the 48-hour period between 00:00 UTC on 23 November and 00:00 UTC on 25 November.

## 4.2 The 21-22 October 2013 rainfall event

On 21-22 October 2013, heavy rainfall hit the medium-to-lower part of the Sturla valley, and marginally the lower Lavagna valley and the town of Chiavari (Fig. 1). The convergence between the cold northern air flow and wet currents from the S generated several storm cells that were channelled from the Liguria Sea in front of Chiavari to the inland valleys, in particular in the Sturla valley (ARPAL, 2013). A very intense and persistent rainstorm stood over a very small area for a

period of 1 to 2 hours, favoured by the orientation of the valley and the local "barrier effect" of the Apennines range. The rainfall event started on the evening of 21 October and ended on the morning of 22 October, 2013. The maximum cumulated rainfall was measured by the Borzone rain gauge – located 1 km N of the area where event landslides were most abundant – which recorded 86.0 mm in 1h and 173.2 mm in 3h, and a cumulated rainfall for the 2-day period that exceeded 188.0 mm (Fig. 3b). At this rain gauge, the 1-hr maximum intensity reached 86.6 mm between 21:00 and 22:00 UTC. The

most intense phase of the event took place between 21:00 and 23:00 UTC on 21 October, and coincided with the maximum peak of the rainfall intensity.

Figure 4b shows the cumulated event rainfall between 18:00 and 24:00 UTC on 21 October 2013. The rainfall field exhibits a NE–SW trend that coincides with the direction of the medium-lower part of the Sturla valley. The maximum cumulated rainfall was recorded along the valley (Borzone, 185.2 mm), with the rainfall totals decreasing gradually towards the main watershed (Giacopiane, 67.0 mm) and the lateral valleys (Cichero, 76.6 mm), up to minimum values at the outlet of the Sturla basin (Panesi, 36.2 mm).

The heavy rainfall event triggered widespread shallow landslides, which caused serious damage to structures and infrastructures (Faccini et al., 2017). Eyewitness evidence provided by local residents confirmed that the shallow landslides occurred in the 30-minute period between 23:00 and 23:30 LT (00:00 and 00:30 UTC), shortly after the most intense phase of the rainfall event, and within 2 to 3 hours from the hourly maximum peak intensity recorded by the Borzone rain gauge in the 24-hour period between 12:00 UTC on 21 October, and 12:00 UTC on 22 October 2013.

### 4.3 The 10-11 November 2014 rainfall event

On 10 November 2014, a very short, intense rainfall hit the town of Chiavari and the inland areas between the lower Lavagna valley and the medium-lower Sturla valley (Fig. 1). The heavy rainfall was caused by the convergence of humid air masses coming from SE and colder northerly currents over the Tigullio Gulf (ARPAL, 2015). Favoured by the local orography, the convective system remained for several hours on the area, producing intense and persistent precipitations. The event began with a light rainfall in the evening of 9 November and ended on 11 November. The maximum cumulated rainfall was recorded by the Panesi rain gauge, located about 2 km E from the centre of the cluster of the event landslides, with 70.4 mm in one hour, 130.6 mm in 3h, 169.4 mm in 6h, and a cumulated rainfall in the 3-day period that exceeded 250.0 mm (Fig. 3c). At the Panesi rain gauge, the 1-hour maximum intensity reached 66 mm between 20:00 and 21:00 UTC. The most intense phase occurred between 17:00 and 23:00 UTC on 10 November, with a maximum intensity peak between 21:00 and 22:00 UTC.

Figure 4c portrays the cumulated event rainfall between 18:00 and 24:00 UTC on 10 November 2014, and shows that the maximum rainfall occurred along a N-S trending line, from the mouth of Entella River (Panesi, 183.2 mm) to the medium-lower Sturla valley (Borzone, 154.2 mm). Large cumulated rainfall amounts also characterize the lateral valleys (Cichero, 135.8 mm), and the upper Graveglia valley (Statale, 130.6 mm), whereas rainfall totals decrease towards W, to the medium (Pian dei Ratti, 82.4 mm) and upper (Ognio, 18.4 mm) Lavagna valley.

The heavy rainfall produced widespread shallow landslides and earth flows, which caused serious damage to structures and infrastructures, and two casualties (Cevasco et al., 2017; Faccini et al., 2015a). Based on eyewitness evidences provided by local inhabitants, and on information obtained from newspaper articles, shallow landslides occurred in the 1-hour period between 20:00 and 21:00 LT (21:00 and 22:00 UTC), during the most intense phase of the rainfall event, and within one

hour from the hourly maximum peak rainfall intensity recorded by the Panesi rain gauge in the 96-hour period between 00:00 UTC on 9 November, and 00:00 UTC on 13 November.

## 5 Results

In the 15-year period 2002–2016, the analysis of the 16 events for which spatial and temporal failure information are known with sufficient accuracy, revealed that rainfall that has resulted in shallow landslides exceeded 70 mm in 1 hour in five cases (26.3%), 100 mm in 3 hours in five cases (26.3%), and 200 mm in 24 hours in four cases (21.0%) (Table 3). The rainfall events affected limited sectors of the Entella catchment, where they triggered 94 known landslides. We note that the five rainfall events with the largest hourly cumulated rainfall exceeding 70 mm (23-24 November 2002, 21-22 October 2013, 10-

11 October 2014, 10-11 November 2014, and 14-15 September 2015; Fig. 1) were responsible for a large number of event landslides (tens to several tens of landslides). These events also caused severe damage to structures and infrastructures, and two casualties. With the exception of the 10-11 November 2014 rainfall event, the three events characterized by 24-hr cumulated rainfall exceeding 200 mm (26 December 2013, 10-11 October 2014, 14 September 2015; Table 1) result in less abundant landslides and less severe damage than we expected.

Visual inspection of Fig. 5 reveals that 2013 and 2014 accounted for a larger number of rainfall events, when compared to the entire period considered, with higher values of the cumulated annual rainfall and of the number of rainy days. In the entire considered period, rainfall events with landslides occurred primarily from September to December (Fig. 6a). This is in agreement with results obtained at the national (Guzzetti, 2000) and regional (Giannecchini, 2006) scales in Italy.

The landslide events occurred in all the three tributaries of the Entella catchment but were most frequent and abundant

between the medium-lower Sturla valley and the lower Lavagna valley (Fig. 6b, Table 4), where the most severe and damaging events occurred on 23-24 November 2002, 21-22 October 2013, and 10-11 November 2014. The events caused shallow landslides, primarily soil-slips (Crosta et al., 2003), slow-moving and rapid to extremely rapid earth flows (Hungr et al., 2001, 2014). The slope failures involved chiefly colluvial deposits of variable thickness, up to four meters (CNR IRPI, 2015), and the superficial layers of the highly weathered clayey bedrock. Large volumes of water mixed with mud, debris

and vegetation were mobilized along the slopes and the thalwegs. Landslides occurred mainly on steep and very steep slopes, in the range of acclivity from 36% to 50% (26) and from 51% to 75% (44), and in vegetated areas, both natural (62) and cultivated (19). Only 13 landslides involved urban areas; however, roads, trails, embankments, buildings, trenches and other man-made structure are often present in the source area of the failures (Fig. 2). In particular, shallow landslides developed mainly within forests and woodlands (48), followed by olive groves (9), transitional woodland/shrubs vegetation (10) and

agricultural lands (10) (Table 5). A number of failures (15) affected terraced slopes, mainly planted with olive groves (11), both still-cultivated and abandoned.

Analysis of the rainfall conditions that have resulted in shallow landslides in the Entella catchment between 2002 and 2016, allowed to define a new event rainfall intensity–event duration, ID empirical rainfall threshold (Fig. 7) for the possible initiation of shallow landslides (Guzzetti et al., 2007; 2008) in the catchment. The black line in Fig. 7a represents the best-fit line $T_{50}$:

$$I = 31.88\,D^{-0.60} \tag{4}$$

The black line in Fig. 7b represents the Gaussian model of the empirical distribution of the $\delta$ values around the central value, $\mu = 0$, measured by the standard deviation, $\sigma = 0.26$. The red curve in Fig. 7c is the 5% ID threshold ($T_5$) for the study area:

$$I = (11.28 \pm 1.04)D^{(-0.60\,\pm 0.02)} \tag{5}$$

where $\Delta\alpha = 1.04$ and $\Delta\gamma = 0.02$ are the standard deviation of the and the $\alpha$, $\gamma$ values and the grey pattern shows a proxy for the uncertainty associated to the threshold, in the range 4 h $<D<$ 170 h. The $T_5$ threshold should Eq. (5) leave 5% of the $(D, I)$ empirical points below the power law curve; in Fig. 7c, two points (5.9%) are below the curve, approaching the number of data points expected below the threshold line. We note that the rainfall conditions that have resulted in landslides in the three severe events analyzed in this work i.e., the 10-11 November 2014, the 21-22 October 2013, and the 23-24 November 2002 events, all plot above the established ID threshold curve. The red vertical line represents the 5% threshold ($T_5$), the grey vertical line portrays the mean of the distribution, corresponding to the 50% threshold ($T_{50}$), and the distance $\delta^*$ between the red and the grey lines is used to calculate the intercept of the 5% threshold curve. The 5% threshold $T_5$, is the curve parallel to the best-fit line $T_{50}$, with intercept $\alpha_5 = \alpha_{50} - \delta^*$ and slope $\beta = 0.60$. Assuming the available set of rainfall events is sufficiently complete and representative for the study area, we can state that the probability of experiencing landslides triggered by rainfall below the obtained threshold is less than 5%. Rainfall events that have resulted in slope failures in the Entella catchment considered for the determination of the thresholds, are in the range of rainfall duration 4 h $< D <$ 169 h and in the range of mean intensity 0.5 mm h$^{-1}$ $< I <$ 4.9 mm h$^{-1}$.

As regard the possible role of the antecedent rainfall conditions in the initiation of rainfall-induced landslides in the Entella catchment, we highlight the results of the relationship between the MAP-normalized cumulated event rainfall and the MAP-normalized antecedent rainfall for the different periods for the Panesi, Borzone and Pian dei Ratti rain gauges i.e., the rain gauges nearest to the central portion of the Entella catchment where the considered rainfall events that have resulted in more abundant, widespread and damaging landslides have occurred. Inspection of Fig. 8 reveals that the scattering of the empirical data is very high, and the event cumulated rainfall and the antecedent rainfall for the different periods that have resulted (red triangles) and have not resulted in landslides, or for which the occurrence of landslides is unknown (black squares), cannot be separated. Significant correlation between the event cumulated rainfall and the antecedent rainfall does not exist, based on the results of a Pearson correlation test (Table 6).

## 6 Discussion

Intense and very intense, geographically limited rainfall events generated by local convective thunderstorms, including the 10-11 November 2004, the 21-22 October 2013, and the 23-24 November 2002 rainfall events (Fig. 6, Table 1), have
affected repeatedly the Entella catchment, triggering shallow landslides, slow-moving and rapid to extremely rapid earth flows. We found that very high intensity rainfall events with cumulated rainfall exceeding 50 mm per hour are frequent in the area, including e.g. 93.0 mm in 1 hour recorded by the Chiavari rain gauge on 24 November 2002, 86.6 mm in 1 hour recorded by the Borzone rain gauge on 22 October 2013, and 66.0 mm in 1 hour recorded by the Panesi gauge on 10 November 2014. An analysis of the available rainfall records between 2002 and 2016 revealed that hourly cumulated rainfall
exceeding 50 mm were recorded 42 times in the 15-year investigation period. In the records, very intense rainfall brought by local convective thunderstorms occurred chiefly in the autumn and winter, and were favoured by the motion of air masses from the Liguria Sea, where warm and moist air from the S and cold air from the N collide, generating convective systems; and by the orographic setting of the Apennines, that form a barrier for the convective cells moving inland from the Liguria Sea (Fig. 1).

Analysis of three, particularly severe rainfall events in the Entella catchment between 2002 and 2016 (Table 1) revealed that the rainfall-induced landslides triggered by the intense or very intense rainfall occurred primarily (i) at the same time, or in a period from one to six hours from the maximum recorded peak rainfall intensity, and (ii) at or near the geographical location where the measured rainfall intensity was highest. Shallow landslides and earth flows initiated chiefly from steep and very steep slopes, primarily in forested and natural areas, and secondarily on cultivated and terraced slopes with different level of
maintenance. Natural vegetated surfaces represent the preferential regions where landslides and erosive processes induced by rainfall easily occur. Generally, agricultural terraced surfaces contribute to maintain the slope stability and the regulation of the rainfall run-off as long as the entire terrace system is well-maintained (Brandolini et al., 2018). We found that both abandoned and still-cultivated, supposedly well-maintained, terraced slopes are involved by landslides: we attribute this fact to insufficient regulation of water run-off and bad preservation of the man-made embankments and dry-retaining stone walls
that nowadays frequently characterized still-cultivated terraces, causing an increase in landslide susceptibility. Roads, trails, embankments and other man-made structures along the slope cause a general reduction of soil permeability and induce negative effects on the natural drainage system and stability, favouring the initiation of shallow landslides . This finding may be useful in catchment-scale landslide early warning systems, and for land planning aimed at reducing landslide risk in the study area.

We compared the new rainfall threshold obtained for the Entella catchment to similar ID empirical threshold curves (Table 7) proposed in the literature for mountain catchments, local areas, or single regions in Italy (Fig. 9). In the range of validity for the threshold (4h < $D$ < 170h), the new threshold for the Entella catchment is lower to significant lower than other thresholds proposed for mountain catchments in Italy. It means that rainfall mean intensity required to trigger shallow landslides in the Entella basin are lower for the same rainfall durations than those expected in other Italian regions on the

average. In particular, the threshold is lower than the curves proposed for (i) the Champeyron mountain catchments in the Susa Valley, Piedmont, NW Italy (Bolley and Olliaro, 1999), (ii) the Moscardo mountain catchment, Friuli, NE Italy (Marchi et al., 2002), and (iii) the Valzangona area, Marche, central Italy (Floris et al., 2004); and it is significantly lower than the threshold curves proposed by Giannecchini (2005) for the Apuane Alps, and by Giannecchini et al. (2012) for the Serchio Basin, Tuscany. The obtained threshold is similar to the curves proposed by Bolley and Olliaro (1999) for the Rho

and Perilleux mountain catchments, in Piedmont. The new threshold for the Entella catchment is also significantly lower than the local threshold proposed by Cancelli and Nova (1985) for the Valtellina valley, in the southern Italian Alps, and the regional thresholds proposed e.g., by (i) Ceriani et al. (1994) for the Lombardy region, northern Italy, (ii) Paronuzzi et al. (1998) for the NE Alps, (iii) Calcaterra (2000) for the Campania region, southern Italy, and by (iv) Aleotti (2004) for the Piedmont region, northern Italy. The new threshold is instead higher than the regional thresholds proposed by Peruccacci et

al. (2017) for physiographic, climatic and meteorological regions in Italy that are similar to the Entella River basin; and in particular, (i) the high mean annual precipitation region (1600mm < MAP < 2000mm), (ii) the Apennines mountain province, and (iii) the region characterized by a temperate climate with dry and hot summer (Csa, in the Köppen-Geiger climate classification system).

We attribute the result firstly to the high landslides susceptibility that characterize the study area  due to the complex

geological, geomorphological and land-use setting, the high-relief topography and the peculiar orographic and meteorological conditions of the region, with a high MAP and the frequent occurrence of convective thunderstorms, short duration and high to very high intensity rainfall events, whose formation is favored by the local orographic setting. Secondarily, we explain the low curve obtained with the possible disturbance in triggering landslides due to the presence of man-made structures at the source areas.  However, comparison with other curves has to take in account the possible

uncertainty due to different rainfall and landslide information analyzed, the different methods used to define the thresholds, or the different criteria adopted to identify the rainfall events and the consequent rainfall duration values. For example, some Authors use information from few particularly severe meteorological events that triggered abundant and destructive landslides, as for the Valtellina (Cancelli and Nova, 1985) or the NE Alps (Paronuzzi et al., 1998), with higher mean intensity values; others consider different type of landslides and involved material, as debris flows for the Apuane Alps

(Giannechini, 2010), with larger rainfall amounts required to fail; some Authors use different criteria to define the rainfall event, i.e., duration and intensity value, as for the Piedmont region (Aleotti, 2004).

As regards the comparison between MAP-normalized cumulated event rainfall and MAP-normalized antecedent rainfall, we conclude that antecedent rainfall conditions do not play a significant role in the initiation of landslides in the Entella catchment.

## 7 Conclusion

Between 2002 and 2016, numerous intense rainfall events hit the Entella River basin, in the Liguria Apennines. The intense to very intense rainfall events were produced by local thunderstorms, and caused abundant and widespread shallow landslides and earth flows, which have resulted in a total of two fatalities and severe damage to public and private structures and the infrastructure.

Analysis of three, particularly severe rainfall events occurred in the Entella catchment in the 15-year investigated period, revealed that the rainfall-induced landslides triggered by the intense or very intense rainfall occurred primarily at the same time, or in a period from one to six hours, from the maximum recorded peak rainfall intensity, at or near the geographical location where the measured rainfall intensity was highest. Using different sources of landslide and rainfall information, we identified 29 rainfall events that have resulted in landslides. Among these 29 events, we used rainfall and landslides information for 16 rainfall events for which the time and location of landslides were known with sufficient geographical and temporal accuracy to define a new rainfall intensity-duration (ID) threshold for the possible initiation of shallow landslides and earth flows in the Entella catchment. Adopting an accepted statistical approach (Brunetti et al., 2010; Peruccacci et al., 2012), we defined the empirical ID threshold corresponding to the 5% exceeding probability, and the associated uncertainty. Comparison of the new rainfall threshold for the Entella catchment to empirical rainfall thresholds proposed in the literature for mountain catchments, local areas, and single regions in Italy revealed that the new threshold is lower than most of the published thresholds, in the range $4h < D < 170h$. We attribute the result mainly to the geological, geomorphological, orographic and meteorological settings of the Entella catchment that highly increase the susceptibility to rainfall-induced shallow landslides, and secondarily to the human disturbance.

To evaluate the possible role of the antecedent rainfall conditions in the initiation of landslides in the Entella catchment, we compared the antecedent rainfall for different periods, from 3 to 30 days, with the cumulated event rainfall that have and have not resulted in landslides between 2002 and 2016 in the study area. The analysis revealed that significant correlations between the event cumulated rainfall and the antecedent rainfall do not exist. Based on this finding, we conclude that the antecedent rainfall conditions did not play a major role in the initiation of shallow landslides in the Entella catchment.

We consider the study presents relevant scientific findings and useful prospects in land planning and landslide risk mitigation: the Entella catchment is the largest tyrrhenian basin in the middle-eastern sector of the Ligurian region – one of the Mediterranean areas with the highest MAP values (more than 4000 mm in 2014) – and it is affected by the low-pressure system called "Genoa Low", that result in very heavy, short duration and local storms. After these severe rainfall events, widespread and damaging ground effects are observed, included rainfall induced shallow landslides and flash floods. A recent study, still in progress, permitted us to identify 20 intense rainfall events resulted in severe ground effects during the period 2001-2017 for each of the catchments of the two tributaries Sturla and Graveglia Stream: 50% of the events affect both the sub-catchments to testify the extremely localized extent of these phenomena.

In order to predict the possible occurrence of surface landslides induced by rainfall, a project for a forecasting and early warning system based on regional rainfall thresholds is under construction by the Ligurian Regional Agency for the Environment Protection (ARPA). The authors hope that the approach adopted, and the results obtained in the Entella

catchment will be useful for the implementation of an early warning system, based on precipitation or forecast measurements, not only at regional scale, but also at level of both major basin and single sub-basin. This consideration arises from the punctual surveys carried out by the authors which have detected as in an area, during the same thunderstorm, many shallow landslides were triggered, while no phenomena were activated in the close area. The violent storms, in fact, often have diameters lower than one kilometre.

Using this method, further rainfall thresholds may be defined for different exceedance probability levels, in order to identify the correspondent rainfall conditions needed to initiate slope failures. Since instability phenomena involve both natural vegetated and agricultural terraced slopes, still-cultivated and abandoned, and are frequently favoured by human disturbance, we highlight the importance of improving land-planning and providing also a log-term strategy in order to reduce rainfall induced landslides risk in the study area, including restoration of abandoned terraces, recovery of the drainage system,

maintenance and management of the wooded surfaces, rigorous monitoring of urbanization.

We further expect that the approach adopted to define new threshold curves for possible rainfall-induced landslides tested in this work and its implementation in a early warning system, can be used in other mountain catchments, in Italy and elsewhere.

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

**Figure 1**: (A) Geography and morphology of the Entella River basin and the surrounding areas. White triangles show location of rain gauges. Dots show locations of shallow landslides for the 23-24 November 2002 (blue), the 21-22 October 2013 (red), and the 10-11 November 2014 (green) rainfall events. (B) Simplified geological map for the Entella River. 1, quaternary deposit; 2, marly limestone, calcareous or clayey marl; 3, shale, shale with interlayered siltstone and limestone; 4, sandstone; 5, marl and silty marl; 6, ophiolite; 7, chaotic complex. (C) Location map.

**Figure 2**: Example of shallow landslides triggered by intense rainfall in the Entella catchment in the 15-year period 2002–2016. (a) Leivi; view of a earth flow that destroyed a building, causing two casualties, on 10 November 2014. (b) Ne; view of a shallow landslide occurred on 4 January 2014. (c) Mezzanego; damage to a road and private property caused by a shallow landslide on 22 October 2013. (d) San Colombano Certenoli; view of a earth flow that damaged a building on 24 November 2002. **S**ee Fig. 1 for location of the landslide sites.

**Figure 3** Hourly rainfall (blue bars) and cumulated event rainfall (red line) for the three considered rainfall events in the Entella catchment (Table 1). (A) 23-24 November 2002 rainfall event measured at the San Martino del Monte rain gauge. (B) 21-22 October 2013 rainfall event measured at the Borzone rain gauge. (C) 9-11 November 2014 rainfall event measured at the Panesi rain gauge. See Fig. 1 for location of the rain gauges.

**Figure 4**. Spatial distribution of the cumulated event rainfall (mm) for the three considered rainfall events in the Entella catchment (Table 1). White triangles show location of the rain gauges used to prepare the maps, also shown in Fig. 1. Coloured contour lines show event cumulated rainfall. (a) 23-24 November 2002. (b) 21-22 October 2013. (c) 9-11 November 2014.

**Figure 5**: Comparison of (a) the total (cumulated) rainfall, and (b) the mean number of rainy days, in the 15-year period 2002–2016 (blue), in 2013 (red) and in 2014 (green), at selected rain gauges in the Entella catchment. Rain gauges: Cc, Chiavari; Gi, Giacopiane Lago; Re, Reppia, Pa, Panesi; Ci, Cichero, Bo, Borzone. See Fig. 1 for the location of the rain gauges.

**Figure 6**: (a) Seasonal and (b) geographical distribution of rainfall events with landslides in the 15-year period 2002–2016 in the Lavagna, Graveglia and Sturla tributary valleys of the Entella catchment. Landslide sites: BR, Borzonasca; CG, Cicagna; CR, Carasco; FM, Favale di Malvaro; LM, Lumarzo; LR, Lorsica; LV, Leivi; MG, Mezzanego; NE, Ne; OR, Orero; SC, San Colombano Certenoli. See Fig. 1 for location of the landslide sites.

**Figure 7**. (a) Rainfall condition (D,I) (dots) that have resulted in shallow landslides and earth flows in the Entella River basin, in the 15-year period 2002–2016 (Table 1). The black line represents the power law best-fit curve to the empirical rainfall (D,I) conditions. Small dots show single landslides and large dots show multiple landslides. (b) Graphic representation of the threshold corresponding to the 5% exceedance probability for the distribution of the empirical data points (D,I). Black line is the Gaussian model fit of the difference, for the distribution of the empirical data points (D,I). Grey vertical line corresponds to the 50% threshold (mean value). Red vertical line corresponds to the 5% threshold. (c) Red line represents the 5% rainfall threshold for the Entella catchment. Coloured dots show rainfall conditions for the 10 -11November 2014 rainfall event (green), the 21-22 October 2013 rainfall event (red), and the 23-24 November 2002 rainfall event (blue). Small dots show single landslides and large dots show multiple landslides. Grey area around shows uncertainty of the threshold.

**Figure 8**. Relationship between the MAP-normalized cumulated event rainfall $E_{MAP}$, and the MAP-normalized antecedent rainfall $A_{(d)MAP}$, for 3-day, 5-day, 15-day and 30-day antecedent periods, for the Panesi, Borzone and Pian dei Ratti rain gauges. See Fig. 1 for location of the rain gauges. Red triangles show rainfall events with shallow landslides or earth flow (Table 1), and black squares show rainfall events that did not trigger landslides or for which the occurrence of landslides is unknown.

**Figure 9**. (A) Comparison between the ID threshold defined for the Entella catchment and threshold curves available in literature for regions, individual catchments, and local areas in Italy (Tab. 6). Plot is in logarithmic coordinates. Dashed black lines show regional thresholds and continuous black lines show local thresholds. Red line (22) shows the rainfall threshold defined in this work for the Entella catchment. Sources: 1, Cancelli and Nova (1985); 2, Ceriani et al. (1994); 3, Paronuzzi et al. (1998); 4-9, Bolley and Olliaro (1999); 10, Calcaterra et al. (2000); 11, Marchi et al. (2002); 12, Aleotti (2004); 13, Floris et al. (2004); 14-15, Giannecchini (2005); 16–18, Gianecchini et al., (2012); 19–21, Peruccacci et al. (2017). (B) Location map shows sites for which the threshold curves were defined: dots are threshold curves for individual catchments and local areas, squares are regional threshold curves.

**Table 1**. Rainfall events that have triggered – mostly shallow – landslides in the Entella River catchment in the 15-year period 2002–2016. In bold, the three events discussed in the paper. Sites: BR, Borzonasca; CG, Cicagna; CR, Carasco; CV, Chiavari; FM, Favale di Malvaro; LM, Lumarzo; LR, Lorsica; LV, Leivi; MG, Mezzanego; NE, Ne; OR, Orero; SC, San Colombano Certenoli. See Fig. 1 for location of the sites. Abundance of landslides: S, single landslide; M, multiple landslides. Damaged element: B, building; I, infrastructure; R, road; S, structure; C, casualty.

| ID | Date | Site | Abundance | Damage |
|----|------|------|-----------|--------|
| **1** | **23-24/11/2002** | **BR, CV, NE, SC** | **M** | **C, S, I** |
| 2 | 31/10-1/11/2003 | BR, NE, SC | M | S, I |
| 3 | 19-20/2/2006 | BR | S | S, I |
| 4 | 21-23/11/2007 | MG | S | I |
| 5 | 11-12/11/2008 | BR, CR, NE | M | S, I |
| 6 | 1/12/2008 | CR | S | S, I |
| 7 | 19-20/1/2009 | NE | M | S, I |
| 8 | 29-30/11/2009 | NE | M | S, I |
| 9 | 8/12/2009 | CR | S | S, I |
| 10 | 22-25/12/2009 | BR, CR, MG, NE, OR | M | S, I |
| 11 | 7-9/5/2010 | NE, FM | M | S, I |
| 12 | 31/10-2/11/2010 | BR, MG, NE | M | S, I |
| 13 | 23-24/12/2010 | MG | M | S, I |
| 14 | 8/6/2011 | NE | M | S, I |
| 15 | 4-5/9/2011 | NE | M | S, I |
| 16 | 25/10/2011 | NE | M | S, I |
| 17 | 5/11/2012 | NE | S | S, I |
| 18 | 8/3/2013 | MG | M | S, I |
| **19** | **21-22/10/2013** | **BR, CR, LV, MG, NE, SC** | **M** | **C, S, I** |
| 20 | 30/10/2013 | CG, FM | M | S, I |
| 21 | 8-9/11/2013 | SC | S | S, I |
| 22 | 26-27/12/2013 | BR, LM, MG, NE | M | S, I |
| 23 | 4/1/2014 | BR, MG, NE, SC | M | S, I |
| 24 | 16-20/1/2014 | BR, CR, LM, MG, NE | M | S, I |
| 25 | 10-11/10/2014 | CG, LR, MG | M | S, I |
| 26 | 3-6/11/2014 | NE | M | S, I |
| **27** | **10-11/11/2014** | **BR, CR, LV, MG, NE, SC** | **M** | **C, S, I** |
| 28 | 14/9/2015 | BR, CR, FM, MG | M | S, I |
| 29 | 9/2/2016 | NE | M | S, I |

**Table 2**. Rain gauges in the Entella River catchment used to analyze the rainfall conditions that have resulted in landslides in the catchment, in the 15-year period 2002–2016 (Fig. 1). Rainfall records are locally incomplete, and measurements are not available for all rain gauges in the period 1999–2001.

| Rain gauge | Elevation | Period | Tributary |
|---|---:|---|---|
| Chiavari Caperana | 6 | 2002–2015 | Entella |
| Panesi | 25 | 1933–2015 | Entella |
| Pian dei Ratti | 70 | 2012–2015 | Lavagna |
| Pian di Soglio | 75 | 1936–2010 | Lavagna |
| San Michele di Borzonasca | 170 | 1924–2007 | Sturla |
| Tigliolo | 293 | 1924–2010 | Sturla |
| San Martino del Monte | 309 | 1919–2004 | Lavagna |
| Borzone | 386 | 2006–2015 | Sturla |
| Ognio | 490 | 2012–2015 | Lavagna |
| Reppia | 530 | 1972–2015 | Graveglia |
| Statale | 570 | 1934–2015 | Graveglia |
| Sella Giassina | 593 | 2012–2015 | Lavagna |
| Cichero | 615 | 2007–2015 | Sturla |
| Croce di Orero | 640 | 2012–2015 | Lavagna |
| Giacopiane Lago | 1030 | 1924–2015 | Sturla |

**Table 3**. Rainfall conditions used to define the rainfall threshold for the possible initiation of shallow landslides in the Entella catchment. Rain gauges (R): Bo, Borzone; Ci, Cichero; Cc, Chiavari Caperana; Cr, Croce di Orero; Gi, Giacopiane Lago; Md, Monte Domenico; Og, Ognio; Pa, Panesi; Ps, Piana di Soglio; Pr, Pian dei Ratti; Re, Reppia; Se, Sella Giassina; Sa, San Michele; Sm, San Martino del Monte; St, Statale; Ti, Tigliolo. See Fig. 1 for location of the rain gauges. $E_{*h}$, maximum cumulated rainfall recorded in the catchment, for different periods (mm). Rainfall measures that have resulted in landslides: E, event cumulated rainfall (mm); D, event rainfall duration (hr); I, event rainfall intensity (mm hr$^{-1}$). Abundance of landslides (AL): S, single landslide; M, multiple landslides.

| ID | Date | Maximum event cumulated rainfall | | | | | Event rainfall | | | | AL |
|----|------|--------|--------|--------|--------|--------|------|----|------|----|----|
| | | $E_{1h}$ | $E_{3h}$ | $E_{6h}$ | $E_{12h}$ | $E_{24h}$ | E | D | I | R | |
| 1 | 24/11/2002 | 79.4 Sa | 97.2 Ps | 104.6 Ps | 106.8 Ps | 178.8 Ti | 157.0 | 20 | 7 9 | Sa | M |
| 2 | 20/2/2006 | 19.0 Sa | 37.6 Ci | 47.0 Sa | 85.6 Sa | 92.8 Sa | 128.4 | 58 | 2.2 | Gi | S |
| 3 | 12/11/2008 | 21.6 Ci | 31.6 Re | 53.0 Re | 62.6 Bo | 107.0 Bo | 74.4 | 50 | 1 5 | Pa | S |
| 4 | 1/12/2008 | 20 Bo | 49 Bo | 77.8 Bo | 97.4 Bo | 81.8 Cr | 63.5 | 95 | 0.7 | Pa | S |
| 5 | 8/12/2009 | 14.6 Re | 33.0 Re | 64.8 Re | 94.2 Re | 138.8 Bo | 66.6 | 18 | 3.7 | Pa | S |
| 6 | 22/12/2009 | 13.8 Cr | 34.6 Cr | 61.8 Cr | 76.6 Cr | 105.4 Cr | 50.8 | 28 | 1.8 | Pa | S |
| 7 | 22/12/2009 | 13.8 Cr | 34.6 Cr | 61.8 Cr | 76.6 Cr | 105.4 Cr | 203.0 | 51 | 4.0 | Cr | S |
| 8 | 23/12/2009 | 25.2 Ci | 49.2 Ci | 78.0 Ci | 101.0 Ci | 120.2 Ci | 157.0 | 47 | 3.3 | Bo | M |
| 9 | 9/5/2010 | 7.8 Ci | 16.6 Cc | 22.0 Cc | 24.4 Cc | 29.6 Cc | 15.2 | 6 | 2.5 | Cr | S |
| 10 | 2/11/2010 | 40.8 Bo | 54.0 Bo | 84.0 Bo | 132.2 Bo | 180.0 Bo | 228.2 | 64 | 3.6 | Bo | S |
| 11 | 22/10/2013 | 86.0 Bo | 173.2 Bo | 186.2 Bo | 187.4 Bo | 188.0 Bo | 188.0 | 20 | 99.4 | Bo | M |
| 12 | 22/10/2013 | 86.0 Bo | 173.2 Bo | 186.2 Bo | 187.4 Bo | 188.0 Bo | 55.6 | 20 | 2.8 | Pr | S |
| 13 | 30/10/2013 | 51.4 Pr | 70.4 Pr | 71.0 Pr | 71.0 Pr | 89.6 Cr | 71.0 | 5 | 14.2 | Pr | S |
| 14 | 30/10/2013 | 51.4 Pr | 70.4 Pr | 71.0 Pr | 71.0 Pr | 89.6 Cr | 27.4 | 4 | 6.9 | Cr | S |
| 15 | 26/12/2013 | 27.4 St | 53.4 St | 100.4 Ci | 166.6 Ci | 228.2 Og | 188.0 | 28 | 6.7 | Re | M |
| 16 | 26/12/2013 | 27.4 St | 53.4 St | 100.4 Ci | 166.6 Ci | 228.2 Og | 312.0 | 169 | 1.9 | Og | M |
| 17 | 26/12/2013 | 27.4 St | 53.4 St | 100.4 Ci | 166.6 Ci | 228.2 Og | 172.2 | 24 | 7.2 | Pr | S |
| 18 | 2/1/2014 | 5.4 Cr | 12.6 Cr | 20.8 Cr | 40.6 Cr | 50.2 Cr | 38.6 | 34 | 0.1 | Gi | S |
| 19 | 4/1/2014 | 15.6 Ci | 36.4 Ci | 64.4 Ci | 110.2 Ci | 117.2 Ci | 152.6 | 74 | 2.1 | Re | S |
| 20 | 4/1/2014 | 15.6 Ci | 36.4 Ci | 64.4 Ci | 110.2 Ci | 117.2 Ci | 127.6 | 72 | 1.8 | Bo | M |
| 21 | 17/1/2014 | 20.8 Ci | 43.2 Cr | 77.0 Pr | 169.6 Ci | 190.6 Ci | 46.2 | 15 | 3.1 | Pa | S |
| 22 | 17/1/2014 | 20.8 Ci | 43.2 Cr | 77.0 Pr | 169.6 Ci | 190.6 Ci | 117.8 | 21 | 5.6 | Bo | S |
| 23 | 17/1/2014 | 20.8 Ci | 43.2 Cr | 77.0 Pr | 169.6 Ci | 190.6 Ci | 165.8 | 24 | 6.9 | Og | M |
| 24 | 10/10/2014 | 72.2 Se | 161.8 Se | 187.8 Se | 215.4 Se | 271.6 Se | 157.8 | 21 | 7.5 | Gi | M |
| 25 | 11/10/2014 | 72.2 Se | 161.8 Se | 187.8 Se | 215.4 Se | 271.6 Se | 90.8 | 5 | 18.2 | Pr | M |
| 26 | 11/10/2014 | 72.2 Se | 161.8 Se | 187.8 Se | 215.4 Se | 271.6 Se | 109.6 | 4 | 27.4 | Cr | M |
| 27 | 10/11/2014 | 70.4 Pa | 129.2 Pa | 147.8 Pa | 152.4 Pa | 181.8 Pa | 185.4 | 26 | 7.1 | Pa | M |
| 28 | 11/11/2014 | 70.4 Pa | 130.6 Pa | 169.4 Pa | 202.4 Pa | 213.8 Pa | 238.0 | 37 | 6.4 | Bo | M |
| 29 | 11/11/2014 | 70.4 Pa | 130.6 Pa | 169.4 Pa | 202.4 Pa | 213.8 Pa | 239.4 | 31 | 7.7 | Pa | M |
| 30 | 13/11/2014 | 70.4 Pa | 130.6 Pa | 169.4 Pa | 202.4 Pa | 215.6 Pa | 280.0 | 81 | 3.5 | Pa | M |
| 31 | 13/9/2015 | 36.2 Pr | 58.0 Og | 58.2 Og | 76.8 St | 78.6 St | 61.2 | 13 | 4.7 | Pa | S |
| 32 | 14/9/2015 | 109.8 Cr | 159.2 Cr | 169.2 Cr | 188.2 Cr | 236.6 Cr | 180.4 | 21 | 8.6 | Gi | S |
| 33 | 14/9/2015 | 109.8 Cr | 159.2 Cr | 169.2 Cr | 188.2 Cr | 236.6 Cr | 182.6 | 16 | 11.4 | Pr | S |
| 34 | 14/9/2015 | 109.8 Cr | 159.2 Cr | 169.2 Cr | 188.2 Cr | 236.6 Cr | 236.6 | 21 | 11.3 | Cr | S |

**Table 4**. Rainfall events that have triggered – mostly shallow – landslides in the central sector of the Entella catchment, from November 2002 to September 2015.

| Date | Sites where event rainfall induced landslides were most abundant |
|---|---|
| 23-24/11/2002 | Borzonasca, San Colombano Certenoli |
| 31/10 - 1/11/2003 | Borzonasca, San Colombano Certenoli |
| 19-20/2/2006 | Borzonasca |
| 21-23/11/2007 | Mezzanego |
| 11-12/11/2008 | Borzonasca |
| 1/12/2008 | Carasco |
| 22-25/12/2009 | Borzonasca, Carasco and Mezzanego |
| 31/10 – 2/11/2010 | Borzonasca, Mezzanego |
| 23-24/12/2010 | Mezzanego |
| 8/3/2013 | Mezzanego |
| 21-22/10/2013 | Borzonasca, Carasco, Leivi, Mezzanego, San Colombano Certenoli |
| 26-27/12/2013 | Borzonasca, Mezzanego |
| 4/1/ 2014 | Borzonasca, Mezzanego, San Colombano Certenoli |
| 16-20/1/2014 | Borzonasca, Carasco, Mezzanego |
| 10-11/10/2014 | Mezzanego |
| 10-11/11/2014 | Borzonasca, Carasco, Leivi, Mezzanego, San Colombano Certenoli |
| 14/9/2015 | Borzonasca, Carasco |

**Table 5.** Comparison between spatial distribution of the landslides use to define the rainfall threshold and the land-use setting in the Entella River catchment.

| Land-use category | Number of landslides |
|---|---|
| Urban fabric | 13 |
| Complex cropland | 4 |
| Olive groves | 7 |
| Abandoned olive groves | 2 |
| Vineyard | 0 |
| Agricultural land with natural vegetation | 6 |
| Forested areas | 48 |
| Natural grassland and/or high pastures | 4 |
| Transitional woodland/shrub | 10 |

**Table 6.** Results of the Pearson test for the correlation between MAP-normalised cumulated rainfall and the MAP-normalised antecedent rainfall, for periods of 3, 5, 15 and 30 days before the rainfall events that have resulted (L) and have not resulted (R) in landslides, for the
Borzone (A), Panesi (B) and Pian dei Ratti (C) rain gauges. **See** Fig. 1 for location of the rain gauges.

|   |   | Test value, r | | | |
|---|---|---|---|---|---|
|   |   | 3-day | 5-day | 15-day | 30-day |
| A | Rainfall events resulted in landslides | 0.34 | 0.05 | -0.15 | -0.19 |
|   | Rainfall events | 0.03 | 0.03 | 0.06 | 0.02 |
| B | Rainfall events resulted in landslides | 0.22 | 0.07 | 0.05 | -0.03 |
|   | Rainfall events | 0.05 | 0.09 | 0.21 | 0.12 |
| C | Rainfall events resulted in landslides | -0.36 | -0.33 | -0.29 | 0.01 |
|   | Rainfall events | 0.26 | 0.31 | 0.28 | 0.32 |

**Table 7**. Intensity-Duration (ID) thresholds for the initiation of landslides in Italy. Rainfall intensity, I (mm hr$^{-1}$); rainfall duration, D (hr). Geographical extent: R, regional; L, local; C, catchment. Area: area where the threshold was defined. Landslides type: A, all types; D, debris flow; S, soil slip; Sh, shallow landslides. Source: 1, Cancelli and Nova (1985); 2, Ceriani et al. (1994); 3, Paronuzzi et al. (1998); 4-9, Bolley and Olliaro (1999); 10, Calcaterra et al. (2000); 11, Marchi et al. (2002); 12, Aleotti (2004); 13, Floris et al. (2004); 14-15, Giannecchini (2005); 16-18, Giannecchini et al. (2012); 19-21, Peruccacci et al. (2017).

| # | Extent | Area | Landslide type | Threshold curve | Duration range (hr) |
|---|--------|------|----------------|-----------------|---------------------|
| 1 | L | Valtellina, Lombardy | S | $I = 44.668D^{-0.78}$ | 1÷1000 |
| 2 | R | Lombardy | A | $I = 20.1D^{-0.55}$ | 1÷1000 |
| 3 | R | NE Alps | D | $I = 47.742D^{-0.507}$ | 0.1÷24 |
| 4 | C | Rho Basin, Susa Valley, Piedmont | D | $I = 9.521D^{-0.4955}$ | 1÷24 |
| 5 | C | Rho Basin, Susa Valley, Piedmont | D | $I = 11.698D^{-0.4783}$ | 1÷24 |
| 6 | C | Perilleux Basin, Piedmont | D | $I = 11.00D^{-0.4459}$ | 1÷24 |
| 7 | C | Perilleux Basin, Piedmont | D | $I = 10.67D^{-0.5043}$ | 1÷24 |
| 8 | C | Champeyron Basin, Piedmont | D | $I = 12.649D^{-0.5324}$ | 1÷24 |
| 9 | C | Champeyron Basin, Piedmont | D | $I = 18.675D^{-0.565}$ | 1÷24 |
| 10 | R | Campania | A | $I = 28.10D^{-0.74}$ | 1÷600 |
| 11 | C | Moscardo Torrent | A | $I = 15D^{-0.70}$ | 1÷30 |
| 12 | R | Piedmont | Sh | $I = 19D^{-0.50}$ | 4÷150 |
| 13 | C | Valzangona, Apennines | A | $I = 18.83D^{-0.59}$ | 24÷3360 |
| 14 | C | Apuane Alps, Tuscany | Sh | $I = 26.871D^{-0.638}$ | 0.1÷35 |
| 15 | C | Apuane Alps, Tuscany | Sh | $I = 38.363D^{-0.743}$ | 0.1÷12 |
| 16 | C | Middle Serchio Basin, Tuscany | Sh | $I = 43.48 D^{-0.74}$ | 2÷70 |
| 17 | C | Middle Serchio Basin, Tuscany | Sh | $I = 43.25 D^{-0.78}$ | 1.5÷80 |
| 18 | C | Middle Serchio Basin, Tuscany | Sh | $I = 41.39 D^{-0.76}$ | 1÷80 |
| 19 | R | High MAP regions | A | $I = 8.9 D^{-0.57}$ | 1÷544 |
| 20 | R | Apennine mountain system | A | $I = 8.6 D^{-0.64}$ | 1÷918 |
| 21 | R | Csa climate regions | A | $I = 8.6 D^{-0.65}$ | 1÷1176 |

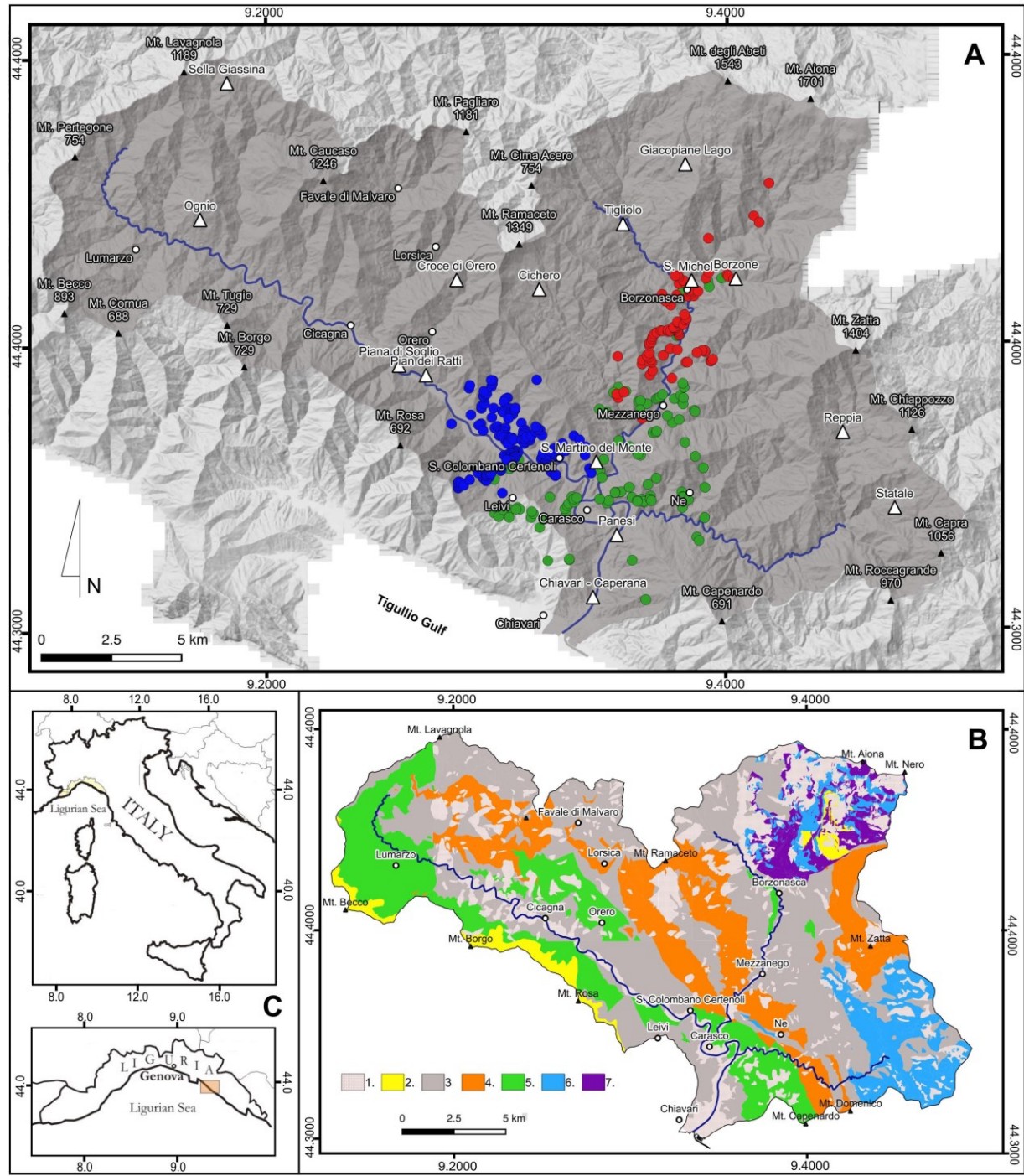

**Figure 1**

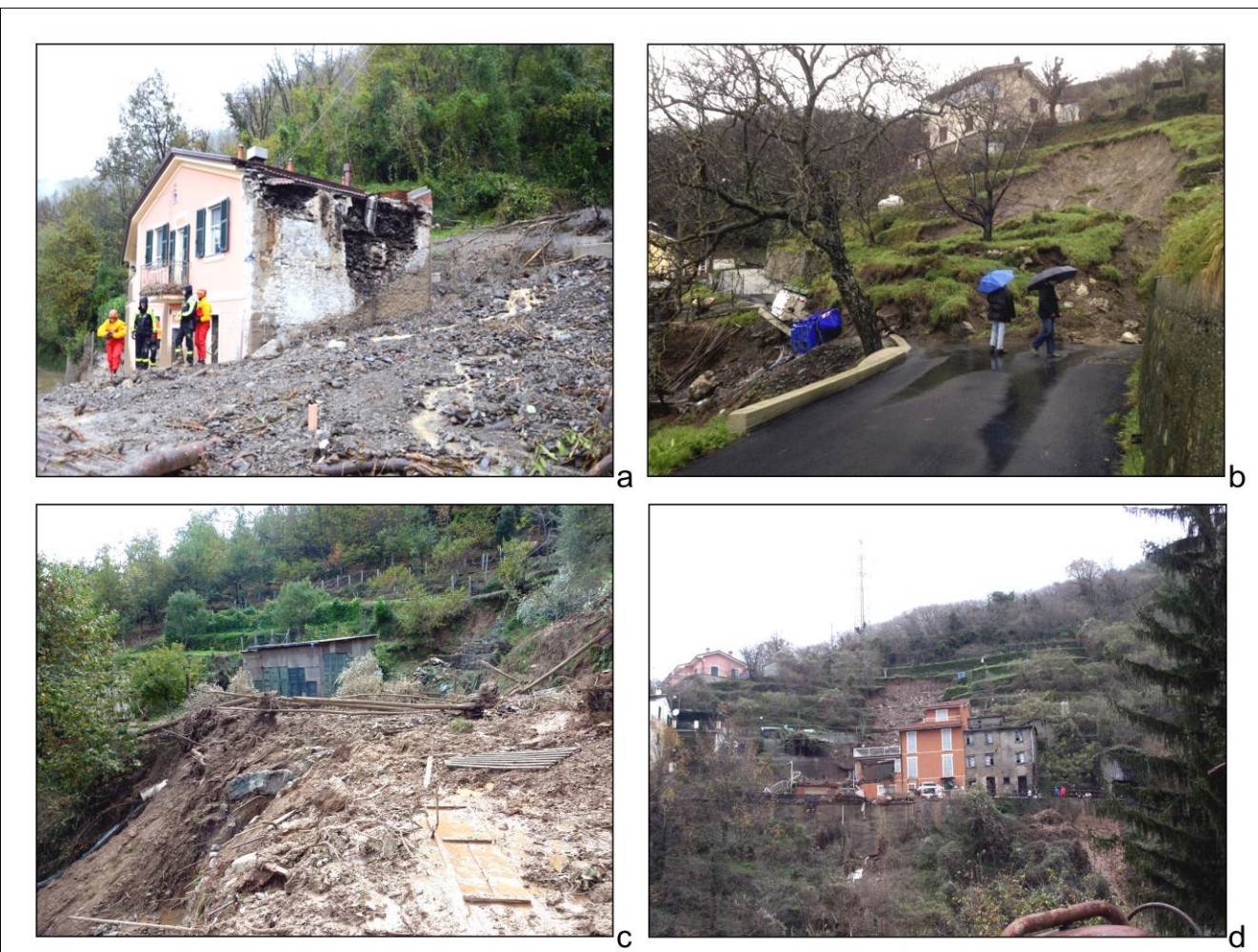

**Figure 2**

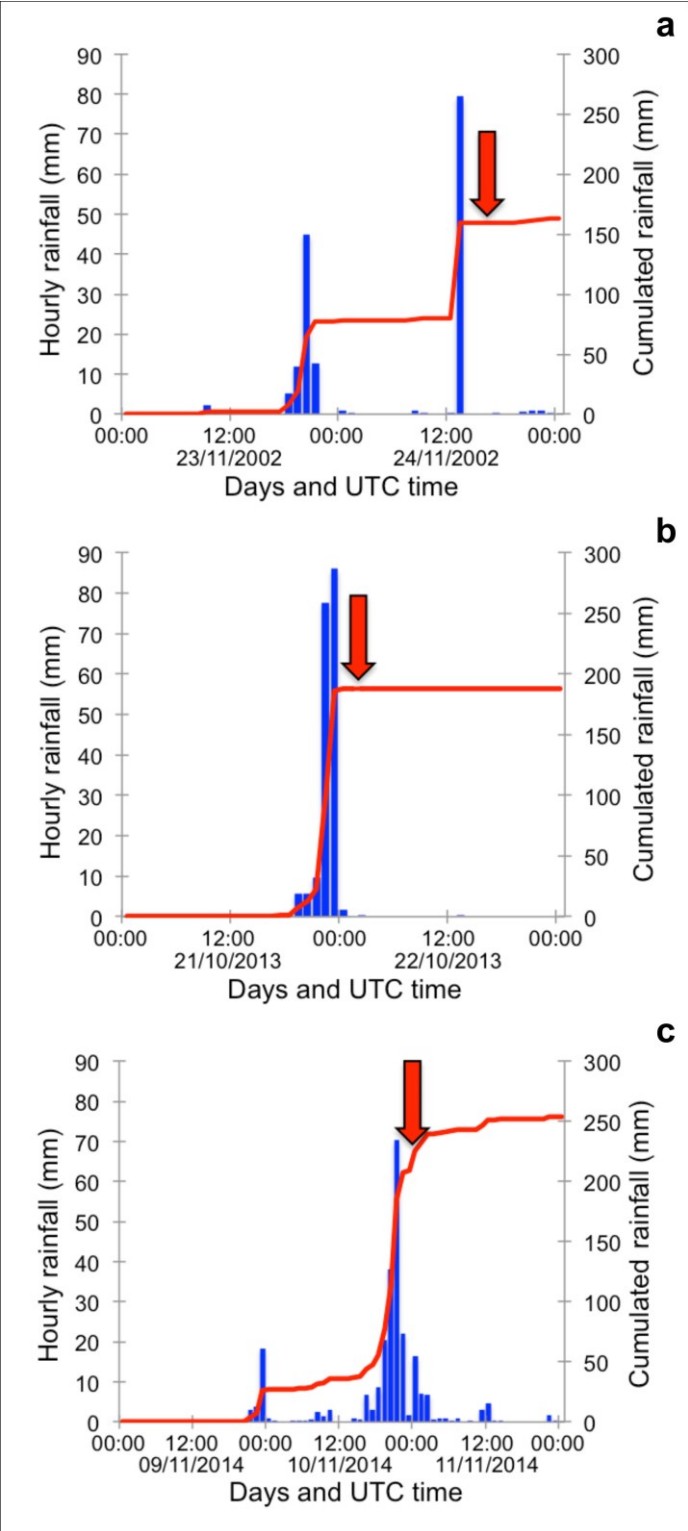

**Figure 3**

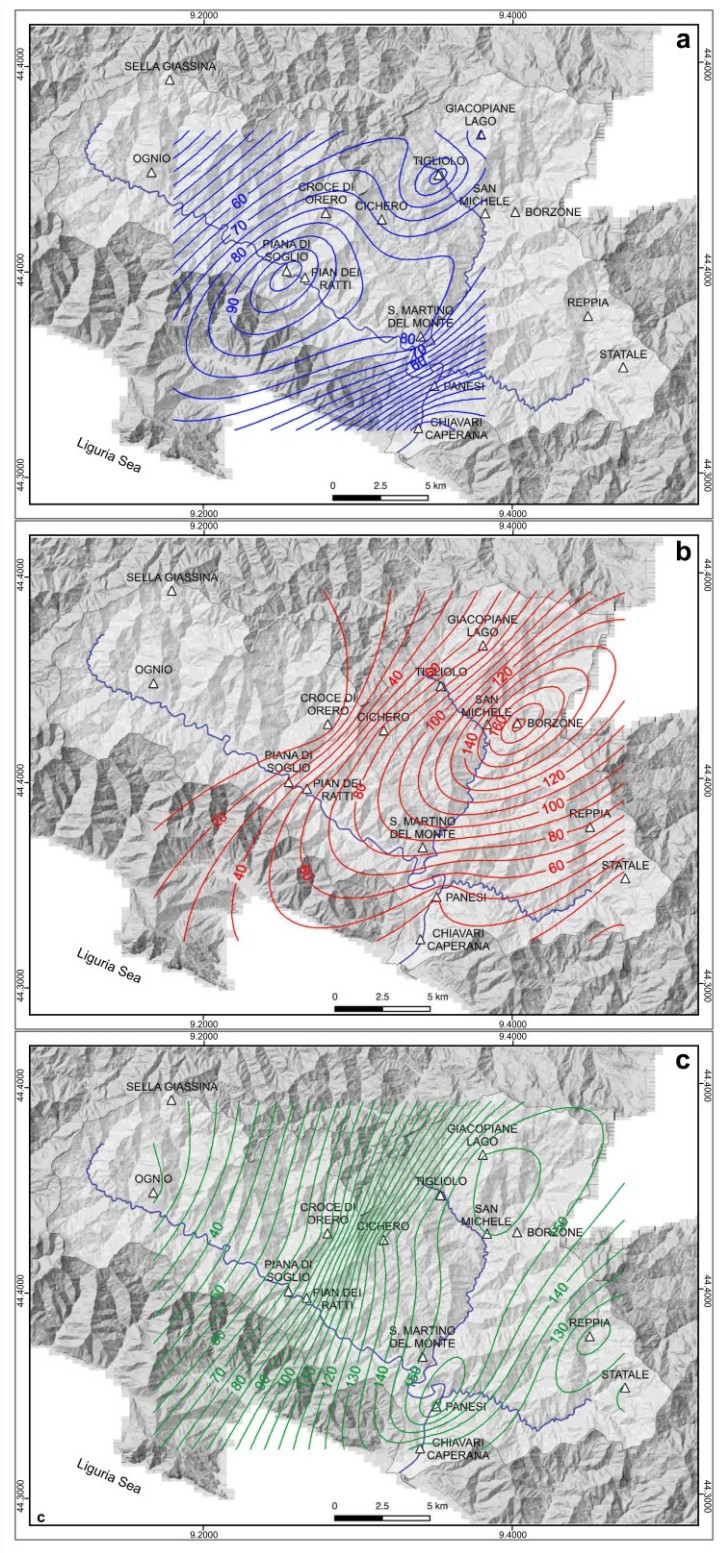

**Figure 4**

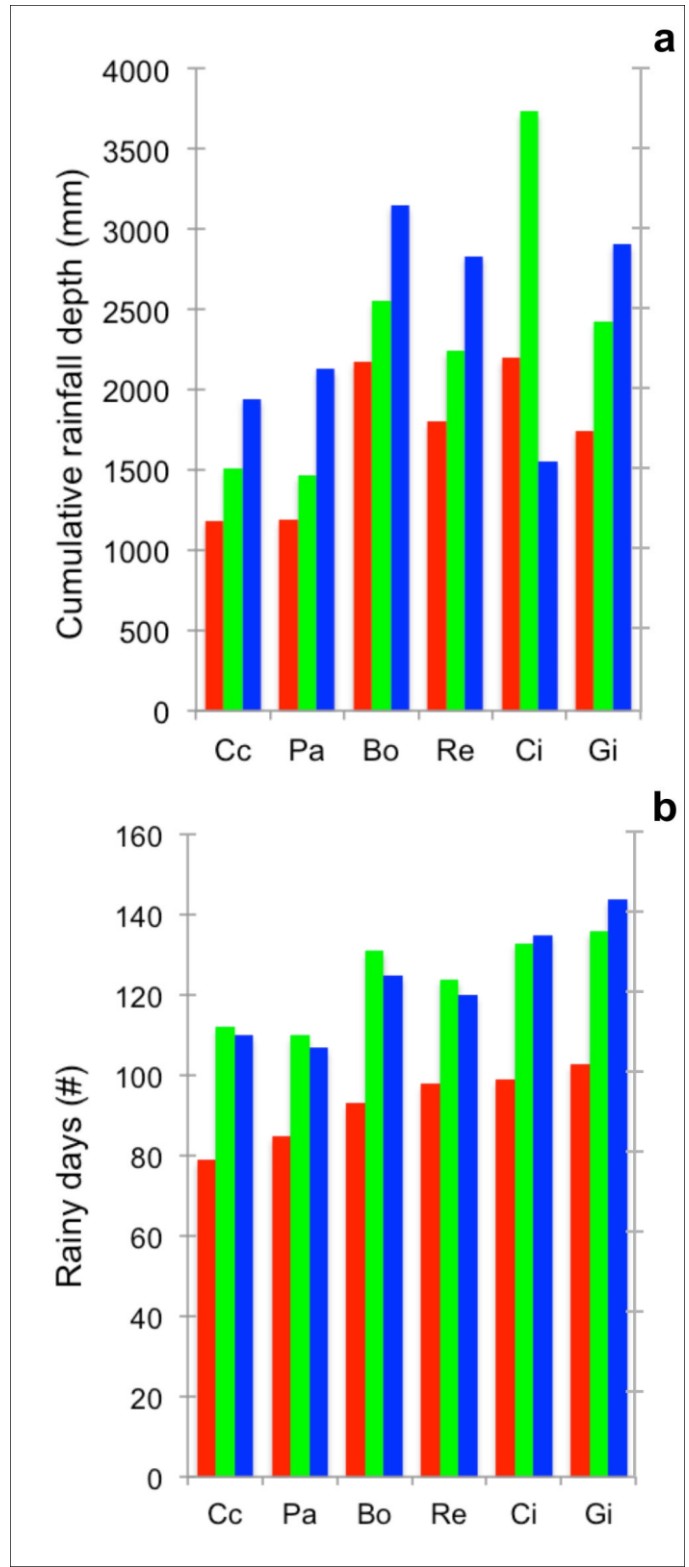

**Figure 5**

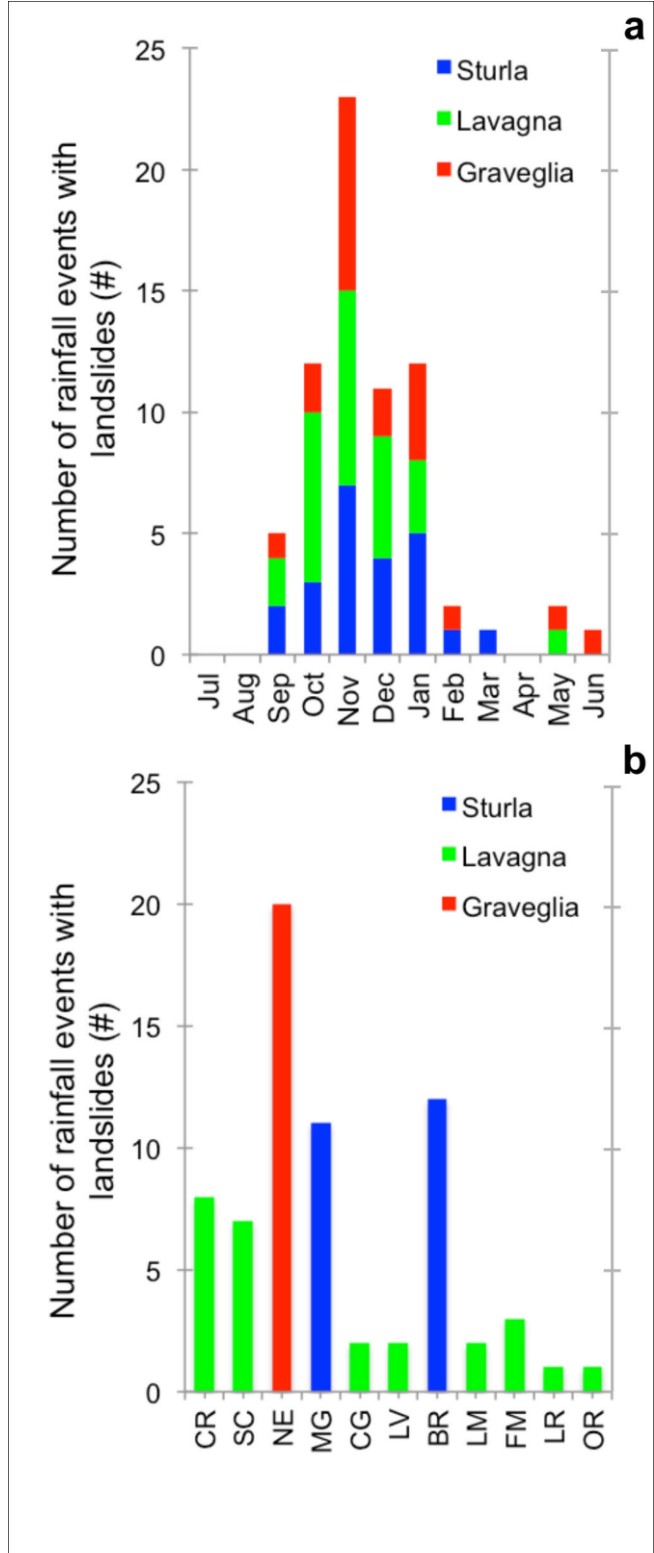

**Figure 6**

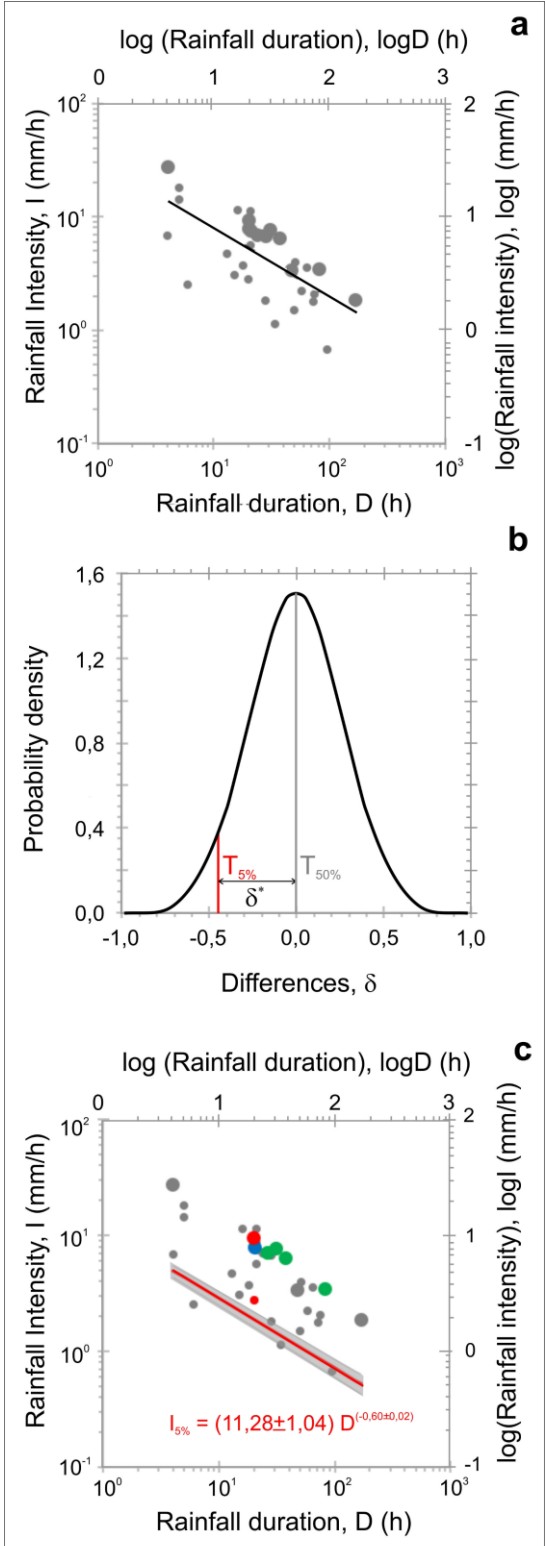

**Figure 7**

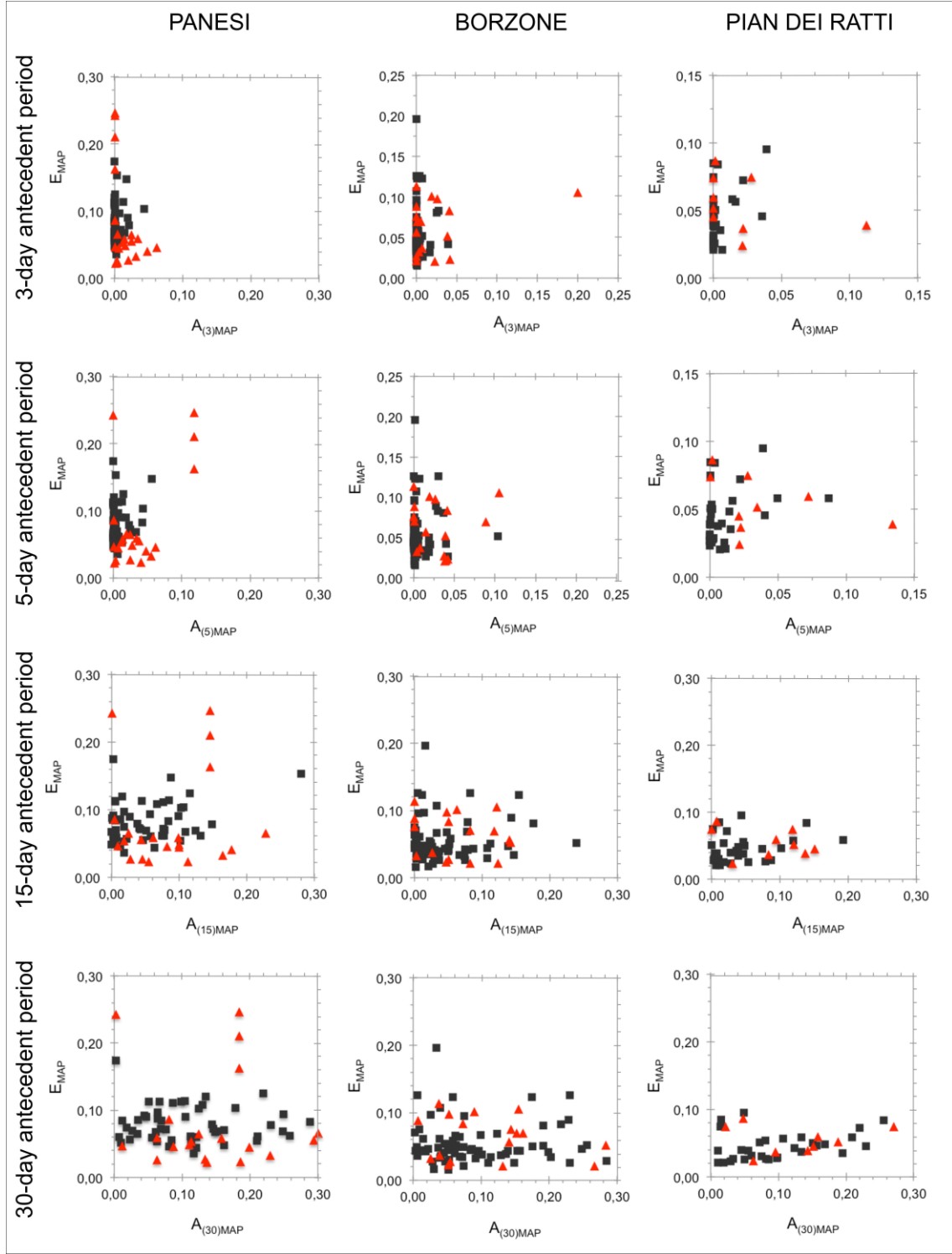

**Figure 8**

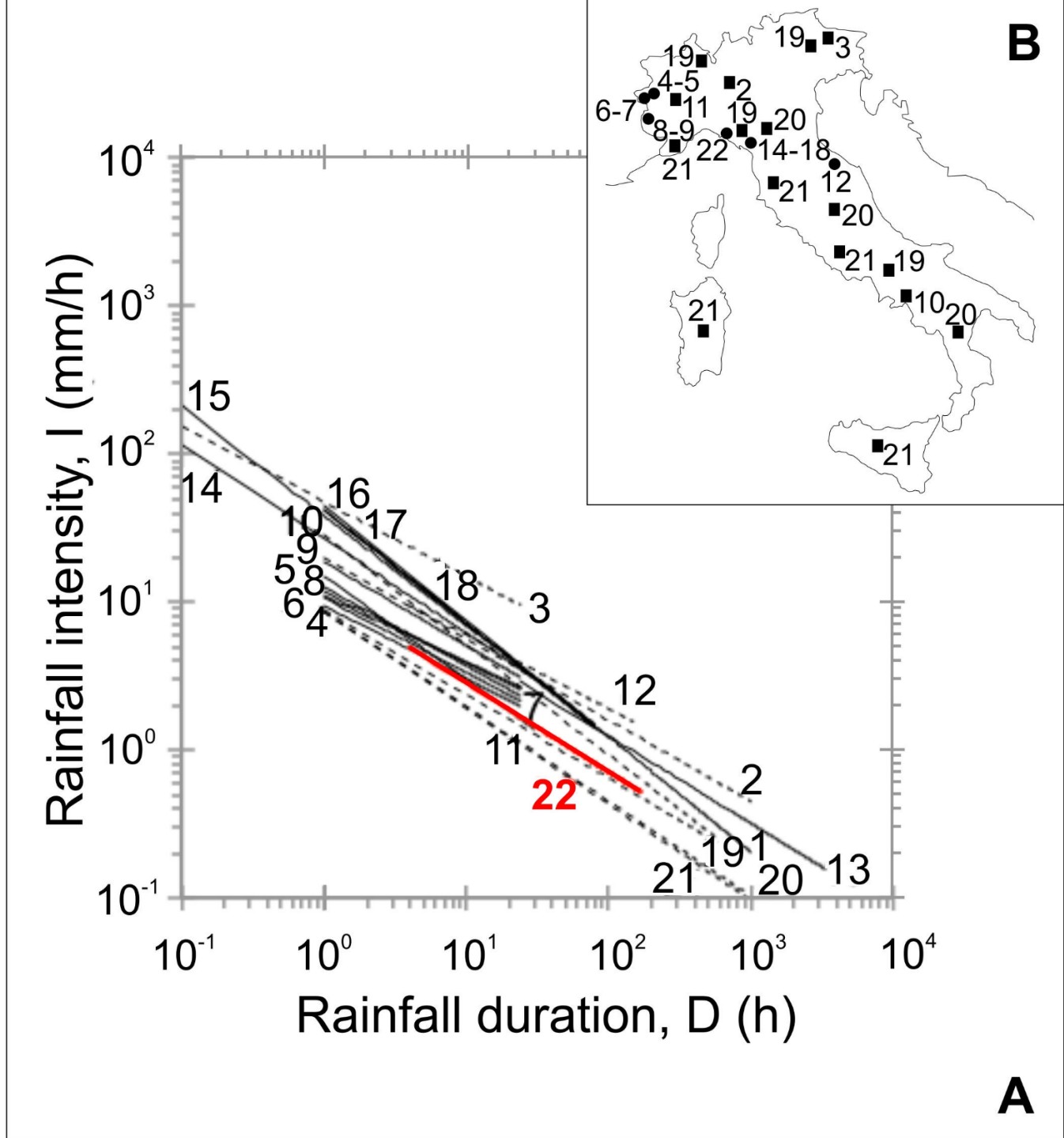

**Figure 9**