# Peer review of "Rainfall events with shallow landslides in the Entella catchment, Liguria, Northern Italy"

_Natural Hazards and Earth System Sciences, 2017_

## Referee Comment (RC1) · Anonymous Referee #1 · 16 Jan 2018

(1) For each rainfall condition, we estimated the cumulated event rainfall, please provide the estimate method more detail. (2) For the landslide inventory, in the text 0f the type, number, and distribution of the event landslides, how to get the landslide occured time, how to determine the the time resolution, for example, which hour did landslide occurr? (3) How many landslide occur in the normal rain peorid (for example, rain in several hours, or one day)? the paper focus on three main rain events? The threhold is the same or not? (4) One geology rock type map shall be help.

---

## Author Comment (AC1) · 24 Jan 2018

(1) For each rainfall condition, we estimated the cumulated event rainfall, please provide the estimate method more detail. We estimated the cumulated event rainfall as the cumulative (total) rainfall measured during the rainfall event, determined measuring the period between the time of the landslide(s), set to coincide with the end-time of the rainfall event, and the time when the rainfall started in the rainfall record, set to coincide with the start-time of the rainfall event, as described in section 5, line numbers 202-207

(2) For the landslide inventory, in the text 0f the type, number, and distribution of the event landslides, how to get the landslide occurred time, how to determine the time resolution, for example, which hour did landslide occur? As described in section 3, we

obtained the landslides information from different sources, which can report the number of soil slips, their location and time of occurrence of the shallow landslides with different geographical and temporal accuracy. When the sources reported accurate information about the time (known or inferred) of initiation of the landslide, we determined the landslide triggering time with an hourly accuracy. For landslides for which only the date was known, we set the time of initiation of the landslide to coincide with the time of the last rainfall measurement of the day in which the slope failure occurred, with a daily accuracy (see section 5).

(3) How many landslide occur in the normal rain period (for example, rain in several hours, or one day)? the paper focus on three main rain events? The threshold is the same or not? Information available from the sources is not always accurate and reliable about the type and the number of the landslides triggered in each rainfall event. In many case, the sources reported indefinite or approximate descriptions, e.g. "a number of landslides" or "multiple slope failures" or simply "landslides". Moreover, for each rainfall event the number of shallow landslides that really occurred cannot be determined exactly. A larger (or much larger) number of landslides may have occurred (including e.g., landslides occurred in forested or remote areas), and were not reported by the sources. As shown in Table 1, we classified the abundance of the landslide as "single" or "multiple"; on the basis of the available information, the exact number of slope failures occurred in a given rainfall period remains unknown.

The paper analyses the 29 rainfall events occurred in the 15-year period between 2002 and 2016 (see Table 1) and focuses on the three main rain events, in terms of ground effects, damage and loss of human lives. As reported in sections 4 and 6, to define the rainfall thresholds we considered the 17 rainfall events (among the 29 listed events) for which the time and location of the slope failures were known with sufficient geographical and temporal accuracy (see Table 3), including the three main events discussed in detail.

(4) One geology rock type map shall be help For a simple and immediate interpretation of the complex geological setting of the study are, we adopted a simplified geo-lithological map, shown in Figure 1B.

Turin, 24 January 2018
* * *

---

## Referee Comment (RC2) · Anonymous Referee #2 · 4 May 2018

My review is composed by two parts. First, I provide some general and specific comments. Second, I answer the questions provided by the journal guidelines.

GENERAL COMMENTS The manuscript is about a case study where three main rainfall events are thoroughly described; after, already published and well established methods are applied. Therefore, the only value of the manuscript is presenting the new case study, while no relevant scientific findings are presented. It seems to me that this is a sufficient reason for rejection, but the Editor will have the final decision on that.

The introduction is poor. It fails to present the existing state of the art and to introduce the advances presented with this manuscript. Moreover, the introduction is centered only on the case of study. The scientific literature is not properly addressed, thus it is not clear the novelty and the improvements conveyed by the work.

References should be completely revised. They are very numerous but they are very biased: almost only Italian authors are present. Some of them with a unnecessary high number of works. Moreover, some of the references are unnecessary or not relevant (e.g. conference proceedings). I suggest to reduce the references and address the recent international literature.

The discussion is not a discussion. It starts with a recap, then it introduces some elaborations (rainfall threshold analysis) that in my opinion should be described in the methodology and in the result section.

The comparison with some literature thresholds is trivial. It is obvious that different sites are represented by different thresholds. I suggest to either cut this descriptive part, or to deeply discuss the reasons of the similarities/differences.

Some of the conclusions are not supported by data and are just speculations: -How can the study be useful for "land planning" and "risk reduction"? You didn't provide a susceptibility map or a hazard assessment. - the "method to define..." is not original work of this manuscript. It is a published and quite well established procedure.

SPECIFIC COMMENTS L21: a reason for the low threshold should be provided

Study area: typical landslides should be characterized (at least typology and size)

L71: why inundations are included in a work about landslides?

L69-81: two sections cannot have the same title.

L89: please, avoid generic terms like "most". How many of them?

L113-177: I don't understand the reason of including detailed event reports in a research paper. This part could be consistently shortened or cut.

L194-198: these are just generic statements. They are not supported by data. It would be interesting to see statistics and numbers. E.g. what's the difference between an abandoned and a maintained terrace? From your text it seems that in both cases they

increase landslide susceptibility. This is a very strange statement: can terraces be abandoned harmlessly?.

L275-281: As I understand, rainfalls have been normalized by the MAP registered by three rain gauges. This is not correct: each rainfall should be normalized by the MAP registered by its own rain gauge.

L296: I think in the text you provided different numbers.

——- ANSWERS TO JOURNAL GUIDELINES Does the paper address relevant scientific and/or technical questions within the scope of NHESS? YES, the topic of rainfall induced landslides is within the scopes of NHESS.

Does the paper present new data and/or novel concepts, ideas, tools, methods or results? NO. Data are unpublished but concepts, ideas, tools, methods and results/conclusions are nothing new.

Are these up to international standards? NO (no relevant original content)

Are the scientific methods and assumptions valid and outlined clearly? YES each Are the results sufficient to support the interpretations and the conclusions? NO (see specific comments)

Does the author reach substantial conclusions? NO

Is the description of the data used, the methods used, the experiments and calculations made, and the results obtained sufficiently complete and accurate to allow their reproduction by fellow scientists (traceability of results)? YES

Does the title clearly and unambiguously reflect the contents of the paper? YES

Does the abstract provide a concise, complete and unambiguous summary of the work done and the results obtained? YES

Are the title and the abstract pertinent, and easy to understand to a wide and diversified

audience? YES

Are mathematical formulae, symbols, abbreviations and units correctly defined and used? If the formulae, symbols or abbreviations are numerous, are there tables or appendixes listing them? YES

Is the size, quality and readability of each figure adequate to the type and quantity of data presented? YES

Does the author give proper credit to previous and/or related work, and does he/she indicate clearly his/her own contribution? YES

Are the number and quality of the references appropriate? NO. References are very numerous but they are very biased: almost only Italian authors are present. Some of them with a unnecessary high number of works. Moreover, some of them are unnecessary or not relevant (e.g. conference proceedings).

Are the references accessible by fellow scientists? NO. I doubt that many conference proceedings could be easily accessed.

Is the overall presentation well structured, clear and easy to understand by a wide and general audience? NO. I have concerns on the manuscript structure (see comments below).

Is the length of the paper adequate, too long or too short? ADEQUATE

Is there any part of the paper (title, abstract, main text, formulae, symbols, figures and their captions, tables, list of references, appendixes) that needs to be clarified, reduced, added, combined, or eliminated? Sections 4.1, 4.2 and 4.3 are basically a short version of three event reports. They distract the reader from the scientific content of the manuscript.

Is the technical language precise and understandable by fellow scientists? YES

Is the English language of good quality, fluent, simple and easy to read and understand

by a wide and diversified audience? YES

Is the amount and quality of supplementary material (if any) appropriate? N.A.

---

## Author Comment (AC2) · 7 Jun 2018

GENERAL COMMENTS

\# The manuscript is about a case study where three main rainfall events are thoroughly described; after, already published and well established methods are applied. Therefore, the only value of the manuscript is presenting the new case study, while no relevant scientific findings are presented.

@ We maintain that the manuscript is well within the aims of NHESS. Undoubtedly, several papers on rainfall thresholds and landslides induced by intense rainfall events in the Mediterranean area, including Italy and Liguria region, are published. As it is, we applied an already published and established method: however, we consider that

implementation of the method in a region as the Entella River basin, most frequently affected by intense and severe meteorological events, can consolidate and increase the scientific value of the method itself. At bottom, we would not have all this great scientific heritage on the rainfall thresholds if we had not considered individual studies on thresholds on specific areas!

\# The introduction is poor. It fails to present the existing state of the art and to introduce the advances presented with this manuscript. Moreover, the introduction is centered only on the case of study. The scientific literature is not properly addressed, thus it is not clear the novelty and the improvements conveyed by the work.

@ In the revised final version of the paper, we shall include a brief description of the state-of-art in order to clarify the improvements of our work.

\# References should be completely revised. They are very numerous but they are very biased: almost only Italian authors are present. Some of them with a unnecessary high number of works. Moreover, some of the references are unnecessary or not relevant (e.g. conference proceedings). I suggest to reduce the references and address the recent international literature.

@ We can improve references, reducing as required. Many Italian papers will be erased and enlarged the international section.

\# The discussion is not a discussion. It starts with a recap, then it introduces some elaborations (rainfall threshold analysis) that in my opinion should be described in the methodology and in the result section.

@ We developed discussions according to a consistent thread: starting from the aim of the paper, we applied a method and described the outcome of our study, therefore we discussed and interpreted results, with reference to the knowledge of the territory investigated and the existing state of the art about the topics treated. We can clarify and refine this section, if necessary.

**The comparison with some literature thresholds is trivial. It is obvious that different sites are represented by different thresholds. I suggest to either cut this descriptive part, or to deeply discuss the reasons of the similarities/differences.**

@ Comparison with similar threshold curves proposed in the literature for mountain catchments, local areas or single regions, with analogous physiographic, climatic and meteorological features, in Italy comparable with the Entella River basin is required to discuss and interpret the outcomes of the study. Many preceding studies have developed this kind of approach (Guzzetti in particular). We can improve this section, deeply discussing the reasons of the similarities and the differences between the threshold obtained for the Entella River Basin and the curves proposed in the literature, if necessary.

**Some of the conclusions are not supported by data and are just speculations: -How can the study be useful for "land planning" and "risk reduction"? You didn't provide a susceptibility map or a hazard assessment. - the "method to define..." is not original work of this manuscript. It is a published and quite well established procedure.**

@ We can better investigate the topic about the practice and the usefulness of our study for land-planning and landslides risk mitigation in terms of early warning system, similarity to the flood risk. Although, we adopted an already published and established procedure to define rainfall thresholds, our work represents a suitable employment and case-study in a Mediterranean area characterized frequently by severe and damaging rainfall events, with important implication in terms of landslide hazard assessment and civil protection.

SPECIFIC COMMENTS

**1L21: a reason for the low threshold should be provided**

We attribute the result to the peculiar orographic and meteorological conditions that characterize the Entella catchment, with a high MAP and the frequent occurrence of convective thunderstorms (born on the Ligurian Sea) whose formation is favored by the

local orographic setting (L273-274 in section 5). Furthermore, the low thresholds can be referable to the fact that several shallow landslides triggered on the slopes on the flanks of the road networks, thus they are favoured also by human activity.

**2 Study area: typical landslides should be characterized (at least typology and size)**

The Entella River basin presents a composite geomorphological setting, and features several processes, landforms and deposits due to gravity: different types of landslide can be identified, from DSDDG (particularly at the head of the valleys) to falls, slides, flows and complex movements, with different states of activity, from active to stabilized. The events analyzed in the present paper, characterized by intense rainfall and short duration, trigger mainly shallow landslides, such as debris flows or soil slips: in terms of damage produced and volume involved, they are characterized by a moderate de-structive capacity and a low magnitude, but they are widely spatial distributed in the catchment area.

**3L71: why inundations are included in a work about landslides?**

The rainfall events that we considered to define the rainfall conditions for the possible initiation of shallow landslides in the Entella River basin induced severe ground effects both on slopes and along the watercourses, including local inundations and flash floods that affected the valley floor of the three main tributaries and the coastal alluvial plain. We reported this detail to highlight the damaging feature of the rainfall events and for completeness of information. Like in other parts of the world, shallow landslides and flash floods along small catchment are strictly connected.

**4L69-81: two sections cannot have the same title.**

This is our mistake. The title of section 3 is "Landslides and rainfall data", whereas the correct title of section 4 is "Rainfall events with landslides".

**5L89: please, avoid generic terms like "most". How many of them?**

For each rainfall event, the number of shallow landslides that really occurred cannot

be determined exactly. Information available from various sources is not always accurate and reliable about the type and the number of the triggered landslides. In many cases, the sources reported indefinite or approximate descriptions e.g., "a number of landslides" or "multiple slope failures", or simply "landslides". As shown in Table 1, we classified the abundance of the landslide as "single" or "multiple". In order to define the thresholds, we used the intensity and duration information related to the first landslide triggered for each rainfall event. In many cases we do not know the exact number of slope failures occurred.

**6 L113-177: I don't understand the reason of including detailed event reports in a research paper. This part could be consistently shortened or cut.**

The three events illustrated in the paper (24 November 2002, 21–22 October 2013, and 10 November 2014) represent the most severe events occurred in the considered period 2002-2015, in terms of abundance of landslides and damage. For the authors it's important to highlight the frequency and recurrence of harmful effects.

**7 L194-198: these are just generic statements. They are not supported by data. It would be interesting to see statistics and numbers. E.g. what's the difference between an abandoned and a maintained terrace? From your text it seems that in both cases they increase landslide susceptibility. This is a very strange statement: can terraces be abandoned harmlessly?**

The aim of our paper is to define an event rainfall intensity – event duration, ID empirical rainfall threshold for the possible initiation of shallow landslides in the Entella River basin. As explains in section 3, we used specific landslide information, including (i) the location and number of the event landslides, (ii) the time of occurrence of the slope failures, and (iii) the consequences of the landslides (i.e., type of damage, casualties). It has been got a difficult work of homogenization: in fact, we obtained the landslide information from different sources, including scientific papers, technical and event reports, damage reports, and catalogues compiled by regional and local authorities, archives

of local municipalities, newspaper articles, and interviews to local inhabitants. For this reason, the available information are not homogenous and have been drawn up by people with different scientific rank, belonging to diverse professions/bodies: therefore, information about type and number of landslides triggered by each rainfall event are not homogenous. In an another paper (currently in progress) we are investigating features of landslides occurred in the catchment area and their correlation with different controlling factors i.e., slope acclivity and aspect, land-use, anthropic disturbances etc.

**8 L275-281: As I understand, rainfalls have been normalized by the MAP registered by three rain gauges. This is not correct: each rainfall should be normalized by the MAP registered by its own rain gauge.**

Each rainfall record was normalized to the MAP of the same raingauge, to investigate the possible role of the antecedent rainfall conditions in the initiation of the rainfall-induced landslides in the Entella catchment (L275-278). Next, we reported the results of the analysis obtained for the three rain gauges of Panesi, Borzone and Pian dei Ratti because they represent the rain gauges nearest to the central portion of the Entella catchment, where the considered rainfall events induced more abundant, widespread and damaging landslides.

**9 L296: I think in the text you provided different numbers.**

In the 2002-2016 period, we have identified 29 rainfall events with landslides. The events are listed in Table 1 (see also section 3, L70). Among these 29 rainfall events, we have information about time and location of landslides with sufficient temporal and geographical accuracy only for 16 events (not 17 as indicated in L296, events were wrongly numbered):

1) 24/11/2002 2) 20/2/2006 3) 12/11/2008 4) 1/12/2008 5) 8/12/2009 6) 22-25/12/2009 7) 7-9/5/2010 8) 2/11/2010 9) 21-22/10/2013 10) 30/10/2013 11) 26-27/12/2013 12) 4/1/2014 13) 16-20/1/2014 14) 10-11/10/2014 15) 10-11/11/2014 16) 14-9-2015

The 34 rainfall intensity-duration conditions listed in Table 3 (see section 4, L82-86 and section 5, L199-208) are related to the landslides induced by the 16 rainfall events mentioned above.

---

## Author Response (AR1)

**Rainfall events with shallow landslides in the Entella catchment (Liguria, Northern Italy)**

Anna Roccati[1], Francesco Faccini[2], Fabio Luino[1], Laura Turconi[1], Fausto Guzzetti[3]

[1] Istituto di Ricerca per la Protezione Idrogeologica, Consiglio Nazionale delle Ricerche, Strada della Cacce 73, 10135 Torino, Italy
[2] Department of Earth, Environment and Life Sciences, University of Genoa, Corso Europa 26, 16132 Genova, Italy
[3] Istituto di Ricerca per la Protezione Idrogeologica, Consiglio Nazionale delle Ricerche, Via della Madonna Alta 126, 06128 Perugia, Italy

**SPECIFIC COMMENTS**

**#1** *L21 in previous paper version (L23-25 in reviewed paper): a reason for the low threshold should be provided*

We attribute the result primarily to the peculiar geological, morphological, land-use and meteorological conditions that characterize the Entella catchment, with a high MAP and the frequent occurrence of convective thunderstorms, short duration and high to very high intensity rainfall events, whose formation is favored by the local orographic setting. Secondarily, as suggest the occurrence of shallow landslides frequently along the road network, we explain the low thresholds with the possible disturbance in triggering landslides due to the presence of man-made structure al the source areas (See Section 6).

**#2** *Study area: typical landslides should be characterized (at least typology and size)*

The Entella River basin presents a composite geomorphological setting and features several processes, landforms and deposits due to gravity: the most important types of landslide can be identified with i) shallow landslides, characterized by very short duration and intense rainfalls, and ii) ancient slow deformations, reactivated during prolonged precipitations. The rainfall events analyzed in the present paper trigger mainly shallow landslides and earth flows: they involved moderate volume of material and are featured by a moderate destructive capacity, but they are widely spatial distributed in the catchment area.

**#3** *L71 in previous paper version: why inundations are included in a work about landslides?*

We included inundations because the rainfall events used to define the rainfall conditions for the possible initiation of shallow landslides in the Entella catchment, resulted in widespread and severe ground effects both on slopes and along the watercourses, including damaging flash floods in the valley floor of the three main tributaries and in the coastal alluvial plain. In the revised paper, we discuss rightly landslides, without mentioning inundations and flash floods (except in the general introduction).

**#4** *L69-81 in previous paper version: two sections cannot have the same title.*

This was our mistake. The title of section 3 was "Landslides and rainfall data", whereas the correct title of section 4 was "Rainfall events with landslides". In the reviewed paper, we have reorganized the work and subdivided into new sections.

**#5** *L89 in previous paper version: please, avoid generic terms like "most". How many of them?*

For each rainfall event, the number of shallow landslides that really occurred cannot be determined exactly. Information available from various sources is not always accurate and reliable about type and number of the triggered landslides. In many cases, the sources report indefinite or approximate descriptions e.g., "a number of landslides" or "multiple slope failures", or simply "landslides". Therefore, we do not know exactly the total amount of slope failures occurred. As shown in Table 1, we classified the abundance of the landslide as "single" or "multiple". Having said that, using the 16 rainfall events for which spatial and temporal information with sufficient accuracy were know, we identified and georeferenced 94 landslides and we estimated a maximum landslide density of 28 landslides/km$^2$ in the Lavagna valley (San Colombano Certenoli) during the 24 November 2002 event (see Section 5, L 258).

**#6** *L113-177 in previous paper version (Section 4, in reviewed paper): I don't understand the reason of including detailed event reports in a research paper. This part could be consistently shortened or cut.*

In the last decades, the Ligurian Region and the Entella catchment were affected by several rainfall events resulting in widespread and damaging landslides. We include detailed reports about the 24 November 2002, 21–22 October 2013, and 10 November 2014 events because they represent the most recent and severe events, in terms of abundance of landslides and damage, occurred in the in the Entella catchment. Furthermore, we focused on these three occurrences because accurate and copious rainfall and landslide information are readily available for events occurred in the 2000s, due to the increasing attention and visibility reserved by mass media to natural phenomena, such as severe rainfall events and rainfall induced landslides, particularly when they cause extensive damage or fatalities, and the ever-increasing consideration towards natural risks by Authorities and local Administrations.

**#7** *L194-198: these are just generic statements. They are not supported by in previous paper version data. It would be interesting to see statistics and numbers. E.g. what's the difference between an abandoned and a maintained terrace? From your text it seems that in both cases they increase landslide susceptibility. This is a very strange statement: can terraces be abandoned harmlessly?*

Numbers and statistics about correlation between shallow landslides and land use in the Entella catchment are now illustrated to support out statements in Section 5. We discuss our findings in the next Section 6, and particularly the influence of the land use conditions (urban areas, natural vegetated and agricultural terraces) on the slope stability during intense rainfall events. We observed that shallow landslides involved both abandoned and still-cultivated terraced slopes: terracing control erosion and slope degradation as long as the entire terrace system is well-maintained, but insufficient regulation of water run-off and a bad preservation of the man-made embankments and dry-retaining walls decrease slope stability in response to high-intensity rainfall. In both cases, abandoned and still-cultivated terraced slopes, they increase landslides susceptibility.

For further evidences about agricultural terraced slopes and their response to rainfall events and landslides susceptibility, see also Crosta et al. (2013) and Brandolini et al. (2018).

**#8** *L275-281 in previous paper version: as I understand, rainfalls have been normalized by the MAP registered by three rain gauges. This is not correct: each rainfall should be normalized by the MAP registered by its own rain gauge.*

Each rainfall record was normalized to the MAP of the same raingauge used to define the D, I threshold (Section 3, L178-183 in reviewed paper). In Section 5, L303-311, we reported the results of the analysis obtained for the three rain gauges of Panesi, Borzone and Pian dei Ratti because they represent the rain gauges nearest to the central portion of the Entella catchment, where the considered rainfall events induced more abundant, widespread and damaging landslides.

**#9** *L296 in previous paper version: I think in the text you provided different numbers.*

In the 2002-2016 period, we have identified 29 rainfall events with landslides. The events are listed in Table 1. Among these 29 rainfall events, we have information about time and location of landslides with sufficient temporal and geographical accuracy only for a subset of 16 events (not 17, events were wrongly numbered):

1) 24/11/2002
2) 20/2/2006
3) 12/11/2008
4) 1/12/2008
5) 8/12/2009
6) 22-25/12/2009
7) 7-9/5/2010
8) 2/11/2010
9) 21-22/10/2013

10)30/10/2013
11)26-27/12/2013
12)4/1/2014
13)16-20/1/2014
14)10-11/10/2014
15)10-11/11/2014
16)14-9-2015

The 34 rainfall intensity-duration conditions listed in Table 3 are related to the landslides induced by the 16 rainfall events mentioned above (==see Section 3 in reviewed paper==)

**GENERAL COMMENTS**

*# The manuscript is about a case study where three main rainfall events are thoroughly described; after, already published and well established methods are applied. Therefore, the only value of the manuscript is presenting the new case study, while no relevant scientific findings are presented.*

Every time a study on rainfall inducing shallow landslides is presented, a new case study is shown: yes, it is true. On the other hand, our paper adds an important piece that was still missing from the great puzzle of knowledge about this phenomenology.
If a journal rejected the "case studies", we would never have had the chance to know the papers of Cancelli & Nova on the Valtellina (1983), by Cannon & Ellen on San Francisco Bay (1985), by Brand et al. on Hong Kong (1984), by Wieczorek on central Santa Cruz (1987), by Reichenbach et al. on Tiber Basin (1998). We think it would have been a big loss, isn't it?
Having said this, we maintain that the manuscript is well within the aims of NHESS. Undoubtedly, several papers on rainfall thresholds and landslides induced by intense rainfall events in the Mediterranean area, including Italy and Liguria region, are published. As it is, we applied an already published and established method: however, we consider that implementation of the method in a region as the Entella River basin, most frequently affected by intense and severe meteorological events, can consolidate and increase the scientific value of the method itself.

*# The introduction is poor. It fails to present the existing state of the art and to introduce the advances presented with this manuscript. Moreover, the introduction is centered only on the case of study. The scientific literature is not properly addressed, thus it is not clear the novelty and the improvements conveyed by the work.*

In the reviewed paper, introduction has been extended: a general first paragraph pointing on the study concerning rainfall thresholds and the existing state of art about investigation of the correlation between rainfalls and landslides occurrence and the major methods adopted to define rainfall curves are now presented. Using an inductive approach, from general to local, the following paragraphs have been entirely rearranged, to point out the novelty and the improvements conveyed by the work. The scientific literature has been extended and revised.

*# References should be completely revised. They are very numerous but they are very biased: almost only Italian authors are present. Some of them with a unnecessary high number of*

*works. Moreover, some of the references are unnecessary or not relevant (e.g. conference proceedings). I suggest to reduce the references and address the recent international literature.*

References have been completely revised: we reduced the unnecessary or not relevant references, and address more recent international literature as suggested.

**# *The discussion is not a discussion. It starts with a recap, then it introduces some elaborations (rainfall threshold analysis) that in my opinion should be described in the methodology and in the result section.***

The paper has been entirely revised and reorganized into the new following sections: i) introduction (Section 1); ii) description of the general setting of the Entella catchment (Section 2); iii) description of the sources and criteria used for the compilation of the catalogue of rainfall events that triggered shallow landslides and the method adopted to define the rainfall thresholds (Section 3) ; iv) focus on the three largest and damaging rainfall events (24 November 2002, 21-22 October 2013 and 10 November 2014) that affected the study area (Section 4); v) rainfall thresholds for the possible initiation of shallow landslides in the Entella catchment and results of the rainfall and landslides analysis (Section 5); vi) discussion of our findings, including a comparison with similar, published curves and the correlation with the land-use and human interference (Section 6) and vii) summary of the results of the study and its possible implementation in a landslide warning system and land planning (Section 7).

**# *The comparison with some literature thresholds is trivial. It is obvious that different sites are represented by different thresholds. I suggest to either cut this descriptive part, or to deeply discuss the reasons of the similarities/differences.***

Comparison with similar threshold curves proposed in the literature for mountain catchments, local areas or single regions, with analogous physiographic, climatic and meteorological features, in Italy comparable with the Entella River basin is required to discuss and interpret the outcomes of the study. In the revised paper, this part has been improved: we deeply discussed the similarities and the differences between our threshold and the curves proposed in the literature in Section 6 L339-374.

**# *Some of the conclusions are not supported by data and are just speculations: -How can the study be useful for "land planning" and "risk reduction"? You didn't provide a susceptibility map or a hazard assessment. - the "method to define..." is not original work of this manuscript. It is a published and quite well established procedure.***

Conclusions have been revised and improved, as suggested. We consider the study presents relevant scientific findings and useful prospects in land planning and landslide risk mitigation. The Entella catchment is the largest tyrrhenian basin in the middle-eastern sector of the Ligurian region – one of the Mediterranean areas with the highest MAP values (more than 4000 mm in 2014) – and it is affected by the low-pressure system called "Genoa Low", that result in very heavy, short duration and local storms. After these severe rainfall events, widespread and damaging ground effects are observed, included rainfall induced shallow landslides and flash

floods. A recent study, still in progress, permitted us to identify 20 intense rainfall events resulted in severe ground effects during the period 2001-2017 for each of the catchments of the two tributaries Sturla and Graveglia Stream: 50% of the events affect both the sub-catchments to testify the extremely localized extent of these phenomena.

In order to forecast the possible initiation of rainfall induced shallow landslides and mitigate the landslides susceptibility and risk, a project for a forecasting and early warning system is under construction by the Ligurian Regional Agency for the Environment Protection (ARPAL). We consider results obtained for the Entella catchment will be useful for the implementation of an early warning system, based on precipitation or forecast measurements, not only at regional scale but also at the level of major basins and their single sub-basin. This consideration arises from the punctual surveys carried out by the authors which have detected as in an area, during the same thunderstorm, many shallow landslides were triggered, while no phenomena were activated in the close area. The violent storms, in fact, often have diameters lower than the kilometer.

In terms of mitigation of landslide risk, since instability phenomena involve both natural vegetated and agricultural terraced slopes, still-cultivated and abandoned, and are frequently favoured by human disturbance, we highlight the importance of improving land-planning and providing also a log-term strategy in order to reduce rainfall induced landslides risk, including restoration of abandoned terraces, recovery of the drainage system, maintenance and management of the wooded surfaces, rigorous monitoring of urbanization.

**Rainfall events with shallow landslides in the Entella catchment, Liguria, Northern Italy**

Anna Roccati[1], Francesco Faccini[2], Fabio Luino[1], Laura Turconi[1], Fausto Guzzetti[3]

[1] Istituto di Ricerca per la Protezione Idrogeologica, Consiglio Nazionale delle Ricerche, Strada delle Cacce 73, 10135 Torino, Italy
[2] Dipartimento di Scienze della Terra, dell'Ambiente e della Vita, Università di Genova, Corso Europa 26, 16132 Genova, Italy
[3] Istituto di Ricerca per la Protezione Idrogeologica, Consiglio Nazionale delle Ricerche, Via Madonna Alta 126, 06128 Perugia, Italy

Dear Editor,

after the good revisions of the referees, the original paper has been completely revised and enhanced in each part.

We would like to underline that this paper can be considered significant for the scientific community since the Entella basin is the largest Ligurian Tyrrhenian basin and it is located in the central-eastern sector of the Ligurian arch, among the Mediterranean regions, with the highest average annual rainfall (over 4000 mm in 2014).

Then this area:

1) is affected by the depression on the Gulf of Genoa, and therefore is characterized by short and intense rainfall and localized in space;

2) presents a unique orographic barrier as represented by a chain that rises up to 1800 m above sea level at about 15 km from the coast;

3) as a result of short and intense rainfall, it has always ground effects consisting of rainfall-induced landslides and flash floods; a recent 'work in progress' research on the Sturla and Graveglia sub-basins has permitted to gather information on 20 intense rainfall events with significant ground effects for the period 2001-2017 for each sub-basin. Only half of these events occur in both sub-basins, demonstrating the extreme localized extension of the cloudburst;

4) for this purpose, the SARF project (regional alert system), i.e. the definition of pluviometric thresholds for the possible triggering of landslides, is under construction by CNR-IRPI and ARPA Liguria (Regional Agency for the Environment)*. The project aims to define empirical rainfall thresholds for the possible triggering of rainwater-induced landslides in Liguria and the conception and development of a prototypal warning system for the prediction of landslides on a regional scale based on thresholds.

The results of the project can be used to predict the possible occurrence of pluvio-induced landslides and can contribute to the mitigation of hazards and landslide risk.

For these reasons, we think that the paper is worth publishing on NHESS.

Best Regards.

Fabio Luino and his colleagues

- http://www.irpi.cnr.it/en/project/previsione-di-frane-indotte-da-piogge-e-zonazione-del-rischio-da-frana-in-italia/

[revised manuscript text omitted]